# Investigating Hallucinations of Time Series Foundation Models through Signal Subspace Analysis

**Yufeng Zou**[†]   **Zijian Wang**[§]   **Diego Klabjan**[‡]   **Han Liu**[†‡♮]
[†]Department of Computer Science, [‡]Department of Industrial Engineering
and Management Sciences, [♮]Department of Statistics and Data Science, Northwestern University
[§]School of Computer Science, The University of Sydney
`yufeng.zou@u.northwestern.edu`
`zwan0998@uni.sydney.edu.au`
`{d-klabjan, hanliu}@northwestern.edu`

## Abstract

Times series foundation models (TSFMs) have emerged as a promising paradigm for time series analyses and forecasting, showing remarkable generalization performance across different domains. Despite the efforts made on hallucinations of foundation models, hallucinations of TSFMs have been underexplored in existing literature. In this paper, we formally define TSFM hallucinations in the zero-shot forecasting setting by examining whether a generated forecast exhibits different dynamics from those of the context. Our study reveals that TSFM hallucinations are associated with the loss of context information in hidden states during forward propagation. As such, we propose a methodology to identify signal subspaces of TSFMs and magnify the information through intervention. Experiments demonstrate that our proposed intervention approach effectively mitigates hallucinations and improves forecasting performance. The signal strength measure computed from signal subspaces shows strong predictive power of hallucinations and forecasting performance of the model. Our work contributes to deeper understanding of TSFM trustworthiness that could foster future research in this direction.

## 1   Introduction

Times series analysis is a major research field that facilitates decision making and scientific inference across a broad range of domains, from energy and weather to economy, transport, and system management. As a key task, time series forecasting has motivated the development of distinct approaches including statistical [6, 37] and deep learning [31, 45] models. Despite competitive performance for specific tasks, these models are typically trained on a single domain, without sufficient capability to generalize to different domains. Inspired by the success of foundation models in fields like natural language processing [1, 7], time series foundation models (TSFMs) have recently emerged as a new paradigm towards universal forecasters [4, 8, 11, 17, 26, 34, 44]. Pretrained on large-scale time series data, TSFMs have shown remarkable few-shot and even zero-shot forecasting performance across numerous domains [3, 19], substantially reducing the need for downstream data. The hidden representations of TSFMs are also useful for downstream tasks through the extraction of context time series information [35].

Yet, the performance of TSFMs is often plagued by hallucinations, as with other foundation models. Broadly referring to the generation of unsupported statements or nonsensical content, hallucinations are attributed to incorrect knowledge or insufficient inference capability of a model [20]. Among various hallucination detection and mitigation approaches proposed, mechanistic interpretability and

39th Conference on Neural Information Processing Systems (NeurIPS 2025).

test-time intervention require no additional training and have demonstrated effectiveness for large language models (LLMs) [23, 27, 36, 48] and large vision-language models (LVLMs) [25, 47].

In zero-shot forecasting, where a TSFM is tasked with generating extrapolations based on the extracted information of context time series such as trends, periodicity, and patterns [19], accurate processing of the context information is essential for generating high-quality forecasts. As such, we study TSFM hallucinations by examining whether a forecast exhibits drastically different dynamics from those of the context, e.g., Figure 1 (a) versus (b). We investigate the underlying mechanisms of TSFM hallucinations through the lens of hidden representations and develop a novel intervention approach to address the identified causes. As far as we know, limited effort has been made on similar research problems in the existing literature. We strive to address these knowledge gaps and contribute to deeper understanding of TSFM trustworthiness, fostering future research in this direction.

We formally define TSFM hallucinations in the zero-shot forecasting setting in §3 and outline the knowledge rules for checking hallucinations in practice. In §4.1, we gain insights on TSFM hallucinations through experimental analyses, where we find that hallucinations are associated with a lack of context information in hidden states during forward propagation. We then propose a method to identify the signal spaces along with a measure (SSAS) to quantify the signal strength of hidden states in §4.2. Built upon these results, we propose a novel intervention approach (SSIM) that mitigates hallucinations by magnifying the signal information of hidden states in §4.3. Extensive experiments in §5.2 demonstrate that while the forecasting performance of TSFMs suffers from hallucinations, our test-time intervention effectively mitigates hallucinations and improves the quality of forecasts, yielding up to 6.62% reduction in the hallucination rate, 93.83% gain in $R^2$, and 13.52% gain in correlation. Moreover, the signal strength measure we propose has strong predictive power of both hallucinations and forecasting performance of TSFMs.

Our main contributions in this work are as follows. (1) We formally define the problem of TSFM hallucinations and outline a set of procedures to check hallucinations. We are the first to systematically study this problem to our best knowledge. (2) We propose a method to identify the signal subspaces of TSFMs along with a measure to quantify the signal strength in TSFM hidden states. (3) We propose a test-time intervention approach to mitigating hallucinations by magnifying the signal information in TSFM hidden states. (4) We conduct extensive experiments on synthetic and real-world datasets across various domains to examine the impact of TSFM hallucinations and demonstrate the effectiveness of our proposed signal strength measure and test-time intervention.

## 2 Related Work

**Times series foundation models.** TSFMs represent a promising paradigm towards generalization across different time series domains and tasks leveraging the knowledge from large-scale pretraining data [4, 8, 11, 17, 26, 34, 44]. TSFMs not only substantially reduce the need for downstream data but have also shown capabilities of producing accurate forecasts even in zero-shot scenarios when forecasting on inputs from unseen domains [3, 19]. While most TSFMs are Transformer based [38] and open sourced, they are diverse in architectural design, tokenization strategies, and pretraining objectives. For instance, Chronos [4] and Chronos-Bolt adopt an encoder-decoder architecture, while TimesFM [11] is decoder-only. Chronos-Bolt and TimesFM truncate the normalized time series inputs into patches, while Chronos discretely quantizes the scaled inputs into a fixed vocabulary. Yet, the forecasting performance of TSFMs suffers from hallucinations when they fail to adequately capture the signal information from inputs, an issue which we study on models from both families.

**Hallucinations.** Hallucination, defined as the generation of unfaithful or nonsensical content, is a fundamental challenge in large foundation models due to their black-box nature [20]. Recent research has examined model hidden representations for hallucination detection and mitigation, based on the hypothesis that factual knowledge is encoded in these states [10, 12, 16]. Studies have identified diagnostic signals in hidden states, showing that outlier or inconsistent activation patterns during generation can indicate potential hallucinations [2, 9, 13, 36, 41]. Complementary approaches focus on hidden state manipulation, demonstrating that truthfulness can be elicited through targeted neuron activation interventions, offering promising directions for reducing hallucinations [23, 25, 36, 46, 47]. We present the first attempt to formally define and systematically study hallucinations of time series foundation models, developing a methodology to detect and mitigate TSFM hallucinations.

**Intervention.** Hidden state intervention has emerged as a powerful technique for controlling neural models' behaviors, as these internal representations serve as causal factors influencing model outputs. Disentangled time series representations have been learned using variational autoencoders [24, 32]. Research in [22, 25, 48] demonstrates effective control over model outputs through activation steering, which identifies linear-interpretable directions in the representation space and guides hidden states along these pathways. Some achieve model output modification by selectively masking specific neuron activations, preventing corresponding generations from occurring [10, 27, 33, 40]. The approach in [43] alters TSFM outputs but does not address specific challenges of time series forecasting. Differently, we propose a novel intervention approach to TSFM hallucination mitigation, which is context adaptive and selectively intervenes model layers.

## 3  Definitions and Preliminaries

Formally, we first describe the forecasts of a time series foundation model and the problem of hallucinations.

**Definition 1** (**TSFM forecasts**). A pretrained time series foundation model, denoted as $\mathcal{M}_\theta$, takes a time series $\boldsymbol{x}_{context} = [x_1, \ldots, x_p]$ of context length $p$ as the input and generates a forecast $\hat{\boldsymbol{x}} = \mathcal{M}_\theta(\boldsymbol{x}_{context}) = [\hat{x}_{p+1}, \ldots, \hat{x}_{p+q}]$ of horizon $q \geq 2$. We call $i$ in $x_i$ or $\hat{x}_{p+i}$ as the $i$-th time series step. For an $L$-layer transformer-based time series foundation model encoder or decoder, we denote the hidden states at different positions of layer $l$ (the outputs of the layer) as a matrix $\boldsymbol{H}^{(l)} = [\boldsymbol{h}_1^{(l)}, \ldots, \boldsymbol{h}_n^{(l)}] \in \mathbb{R}^{n \times d}$, where $d$ is the dimension of hidden states.

**Definition 2** (**Time series forecasting hallucinations**). Suppose for a time series $\boldsymbol{x}_{full} = [x_1, \ldots, x_T]$, a knowledge set $\mathbb{K}$ can be inferred from a partial time series $\boldsymbol{x}_{context} = [x_i, \ldots, x_j]$, $1 \leq i < j < T$ for any $i$ and $j$. The knowledge set $\mathbb{K}$ comprises time-dependent knowledge rules $r$ that hold true for $\boldsymbol{x}_{full}$, i.e., $r(x_i, i) = 1$ for all $1 \leq i \leq T$, or $r(\boldsymbol{x}_{full}) = 1$ in short. In zero-shot time series forecasting, we consider a hallucination to be a forecast that does not conform to the knowledge rules inferred from the context time series. We define the set of hallucinations as $Hallu(\boldsymbol{x}_{context}) = \{\hat{\boldsymbol{x}} : \bigwedge_{r \in \mathbb{K}} r(\hat{\boldsymbol{x}}) = 0\}$.

The goal of hallucination detection is to define a score function $f$ that discriminates hallucinated forecasts of the model, such that for any $\hat{\boldsymbol{x}} = \mathcal{M}_\theta(\boldsymbol{x}_{context})$, we have $f(\boldsymbol{x}_{context}, \hat{\boldsymbol{x}}, \theta) > \tau$ if an only if $\hat{\boldsymbol{x}} \in Hallu(\boldsymbol{x}_{context})$. We mitigate hallucinations through test-time intervention on hidden states so that with the intervention operation $\mathcal{I}$, we obtain non-hallucinated forecasts $\mathcal{M}_{\theta, \mathcal{I}}(\boldsymbol{x}_{context}) \notin Hallu(\boldsymbol{x}_{context})$.

In practice, we sequentially extract a set of knowledge rules $\mathbb{K} = \{r_1, \ldots, r_n\}$ from the context time series to check whether a forecast is hallucinated.

**Trend.** The trend rule checks whether the trend of the forecast conforms to those of the context. We perform ordinary least-square (OLS) regression on $\hat{\boldsymbol{x}}$ and take the first-degree coefficient $\hat{c}$ as the trend if it is significant with the $p$-value $< 0.01$. We then perform OLS on rolling windows of $\boldsymbol{x}_{context}$ and take significant trends $[c_1, \ldots, c_n]$. With the relative difference between trends computed as $diff(c, \hat{c}) = \left| \frac{\hat{c}}{c} - 1 \right|$, the trend rule is satisfied iff the minimum relative difference $\min_i diff(c_i, \hat{c}) < \delta$, or neither the forecast nor the context exhibits a significant trend.

**Frequency.** The frequency rule checks whether the spectral density of the forecast conforms to those of the context. After removing the trend, we compute the spectral densities $[\boldsymbol{f}_1, \ldots, \boldsymbol{f}_n]$ of rolling windows on $\boldsymbol{x}_{context}$ using short-time Fourier transform (STFT) [18] and also the spectral density $\hat{\boldsymbol{f}}$ of $\hat{\boldsymbol{x}}$. With the Jaccard distance between spectral densities computed as $\mathcal{D}(\boldsymbol{f}, \hat{\boldsymbol{f}}) = 1 - \frac{\sum_i \min\{f_i, \hat{f}_i\}}{\sum_i \max\{f_i, \hat{f}_i\}}$, the frequency rule is satisfied iff the minimum Jaccard distance $\min_j \mathcal{D}(\boldsymbol{f}_j, \hat{\boldsymbol{f}}) < \delta$.

**Pattern.** The pattern rule checks whether the pattern of the forecast is close to those of the context. After removing the trend, we compute the relative absolute errors between the forecast and rolling windows $[\boldsymbol{w}_1, \ldots, \boldsymbol{w}_n]$ on $\boldsymbol{x}_{context}$. With the relative absolute error computed as $RAE(\boldsymbol{x}, \hat{\boldsymbol{x}}) = \frac{\sum_i |x_i - \hat{x}_i|}{\sum_i |x_i - \bar{x}|}$, the pattern rule is satisfied iff the minimum relative error $\min_j RAE(\boldsymbol{w}_j, \hat{\boldsymbol{x}}) < \delta$.

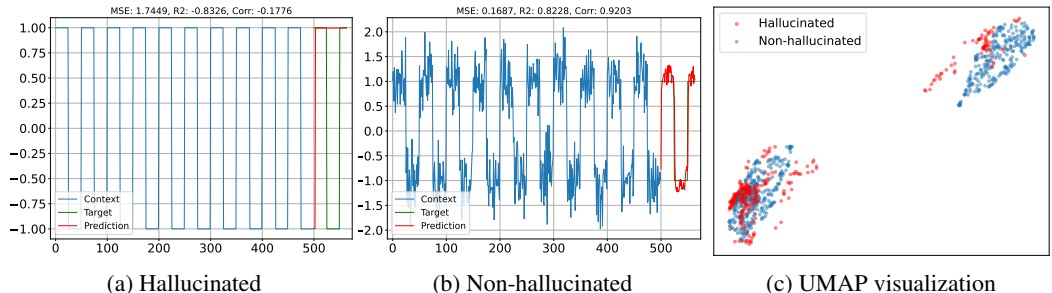

Figure 1: (a) (b) Examples of hallucinated and non-hallucinated forecasts by `Chronos`. (c) UMAP visualization of hidden states at the final model layer for both examples.

**ARMA.** The ARMA rule checks whether the ARMA dynamics of the forecast conform to those of the context, which complements the pattern rule since a time series that exhibits strong ARMA dynamics may not have distinct patterns. After removing the trend, we fit a first-order Autoregressive moving average (ARMA) model [6] on $x_{context}$ and take the AR and MA coefficients $\phi$ and $\psi$ if both are significant with $p$-values less than $0.01$. Let $\hat{\phi}$ and $\hat{\psi}$ be the first-order ARMA coefficients on $\hat{x}$, the ARMA rule is satisfied iff the relative differences $\left|\frac{\hat{\phi}}{\phi} - 1\right| < \delta$ and $\left|\frac{\hat{\psi}}{\psi} - 1\right| < \delta$.[1]

A TSFM forecast that violates either the trend, frequency, or both the pattern and ARMA rules is considered to be hallucinated, since it is ungrounded in the context time series information. Further details are provided in Appendices A and D.3.

## 4 Methodology

To understand the cause of TSFM hallucinations, we first gain insights from observations, provide intuitive explanations, and then perform experimental analyses to justify our claims. Afterwards, we propose a signal strength measure to help detect hallucinations. Finally, we develop a novel test-time intervention approach that mitigates hallucinations by addressing the identified causes.[2]

### 4.1 Observations and Findings

We begin with a brief case analysis. Figure 1 (a) presents a hallucination example where a TSFM fails to generate a forecast consistent with the context time series. The 2-dimensional UMAP [28] visualization of hidden states at the final layer in Figure 1 (c) shows irregular patterns, with a high mean pairwise cosine similarity at $0.8763$ and a low mean activation std across positions at $11.0570$ (we observe two clusters since the context time series is binary). We speculate that the forecast failure is caused by the loss of context information in hidden states during forward propagation. We find that injecting a small amount of random perturbation to the context time series with Gaussian noise alleviates such information loss, as shown in Figure 1 (b). We observe that the hidden states become more evenly distributed in each cluster of Figure 1 (c), with the mean pairwise cosine similarity substantially reduced to $0.6338$ and the mean activation std raised to $14.5772$.

To analyze the effects of context signal and noise on a TSFM when no hallucination occurs, we collect the activations over 10 random perturbations to context signal with Gaussian noise of varying magnitudes. From Figures 2 (a)(b), we observe that hidden state activations exhibit the greatest variance across the positions of a layer with clean signal. As the signal mixes with greater noise, despite the increase of input variance, hidden state activations become less variant. The decline in mean activation variance with noise is more salient at higher layers, suggesting that a TSFM incrementally extracts signals and reduces noises from the input at each layer.

---

[1]This rule is not based on rolling windows for computational efficiency.

[2]We check hallucinations based on the rules defined in §3 with parameters specified in §5.1. The results and observations are not sensitive to these parameters.

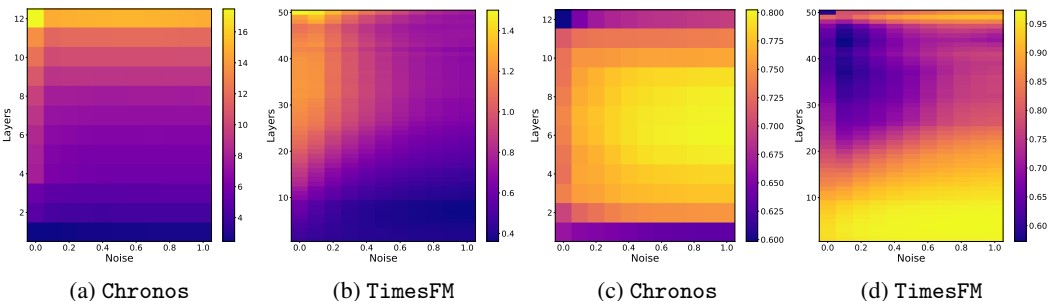

Figure 2: (a) (b) The mean standard deviations of hidden state activations across the positions of a layer under varying noise magnitudes. (c) (d) The mean pairwise cosine similarities of hidden states across the positions of a layer. The $x$-axes represent the standard deviation of Gaussian noise.

Based on this, we posit that the hidden state space $\mathbb{H}^{(l)}$ at each TSFM layer can be decomposed into signal and noise subspaces $\mathbb{H}^{(l)} = \mathbb{S}^{(l)} \oplus \mathbb{N}^{(l)} \subset \mathbb{R}^d$, respectively handling the signals and noises of the input. Decomposition based on null spaces and similar concepts have been considered in the past [14, 21, 30], but herein we introduce the idea of signal and noise. In forward model propagation $\boldsymbol{H}^{(1)} \to \ldots \to \boldsymbol{H}^{(L)}$, the signal components of a hidden state $\Pi_{\mathbb{S}^{(l)}} \boldsymbol{h}^{(l)}$ are further processed by subsequent layers, while the noise components $\Pi_{\mathbb{N}^{(l)}} \boldsymbol{h}^{(l)}$ are repressed and eventually removed. Since the signal components are more variant and dissimilar across hidden state positions than the noise components, the hidden states would exhibit greater distinctiveness across positions with strong signal presence at a layer (Figures 2 (c)(d)).

Reporting Figures 1 (a)(c) of TSFM hallucinations, the inactivity of signal subspaces of the model leads to homogeneous hidden states across positions. In such case, a proper amount of input random perturbation injects variances that help restore the activity level of signal subspaces and facilitate the propagation of context signal information. Nonetheless, it is hard to determine the optimal amount of perturbation, since too much perturbation obscures the input signal and degrades forecast quality. Moreover, a single perturbation is not robust [25], while performing multiple perturbations hampers efficiency. Our goal is to magnify the signal information of hidden states through test-time intervention, which would enable us to mitigate hallucinations in a controllable and efficient manner.

## 4.2 Signal Subspace Identification

Next, we develop a novel method to identify the signal subspaces in TSFM layers and provide empirical analyses. We aim to identify a set of hidden state neurons that are most active to context signals by examining the variance of activations across hidden state positions, enlightened by the associations between the activation variance and signal strength we observed in the previous section. Let $\boldsymbol{H}_{i,j}^{(l)}(\boldsymbol{x})$ be the activation of neuron $j$ at position $i$ of layer $l$ given input $\boldsymbol{x}$ and $\bar{\boldsymbol{h}}_j^{(l)}(\boldsymbol{x})$ be the mean activation of neuron $j$ with respect to $i$. We compute the activity score of neuron $j$ as

$$\mathcal{A}^{(l)}(j \mid \boldsymbol{x}) = \sqrt{\frac{1}{n} \sum_i \left( \boldsymbol{H}_{i,j}^{(l)}(\boldsymbol{x}) - \bar{\boldsymbol{h}}_j^{(l)}(\boldsymbol{x}) \right)^2} . \tag{1}$$

The neuron activity measure we propose is more nuanced compared with the magnitude of neuron activations used in prior works [39, 40], as it not only reflects the overall magnitude but also measures the deviation of neuron activations across time series steps.

To measure neuron activity in the presence of signals, we collect the activity scores on a synthetic dataset comprising common waveforms that is described in §5.1. We also vary the magnitude of noises injected to the context signals and initialize them with different random seeds for robustness. With $\mathcal{D}_{signal}$ denoting the set of synthetic time series inputs where no hallucination occurs, we consider neurons with the activity score consistently top ranked across the samples as candidate signal neurons, i.e., $Cand(l) = \bigcap_{\boldsymbol{x} \in \mathcal{D}_{signal}} \{j \mid \mathcal{A}^{(l)}(j \mid \boldsymbol{x}) < q_\epsilon\}$, where $q_\epsilon$ is the $\epsilon$ quantile of the underlying sequence (in this case $\mathcal{A}^{(l)}(\cdot \mid \boldsymbol{x})$). We compute the signal activity score of each neuron as the sample mean $\mathcal{A}_{signal}^{(l)}(j) = \frac{1}{|\mathcal{D}_{signal}|} \sum_{\boldsymbol{x} \in \mathcal{D}_{signal}} \mathcal{A}^{(l)}(j \mid \boldsymbol{x})$.

Since hidden state neurons may fulfill multiple roles [5, 42, 47], e.g., signal processing and noise removal in concurrence, we aim to identify neurons that are primarily responsible for signal processing. To this end, we collect the activity scores $\mathcal{A}_{noise}^{(l)}$ using Gaussian noises as the input in the same way and compute the contrastive neuron activity score between signal and noise inputs:

$$\mathcal{A}_{contrastive}^{(l)}(j) = \mathcal{A}_{signal}^{(l)}(j) - \mathcal{A}_{noise}^{(l)}(j) \ . \tag{2}$$

For each model layer $l \in \{1, \ldots, L\}$, we pick candidate neurons with top-ranked contrastive activity scores as the signal neurons, i.e., $Sig(l) = Cand(l) \cap \{j \mid \mathcal{A}_{contrastive}^{(l)}(j) < q_\epsilon\}$. Figure 3 displays the distributions of contrastive neuron activity scores across model layers. We note that at each layer only a small proportion of neurons are exclusively sensitive to signal or noise. Moreover, greater contrastive scores are observed at higher layers, where neurons are more specialized.

Ranking signal neurons by contrastive activity score, we select the top signal neuron's activity score at the final layer as a strength measure of signal information the model has processed, i.e., $\mathcal{A}^{(L)}(j \mid \boldsymbol{x})$ with $j = \arg\max_{j' \in Sig(L)} \mathcal{A}_{contrastive}^{(L)}(j')$. The final layer is selected as it shows the greatest contrastive scores between signal and noise inputs (Figure 3). We call the proposed measure Signal Subspace Activity Score (SSAS) and will verify its efficacy of hallucination detection and forecasting

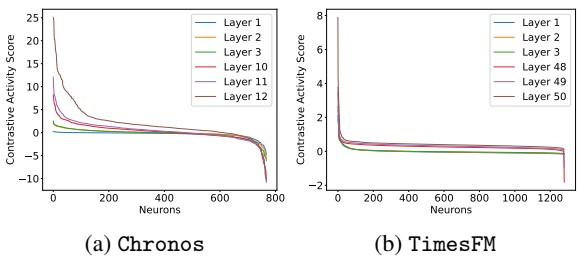

(a) Chronos        (b) TimesFM

Figure 3: Ranked contrastive neuron activity scores across model layers.

performance prediction in §5.2. With this, we claim that a TSFM implicitly expresses in the hidden state subspaces how much signal information it is able to capture from the context.

## 4.3 Signal Subspace Intervention

Built upon previous results, we propose a Center-Project-Scale (CPS) intervention operation to mitigate hallucinations by magnifying hidden state signal information. Setting $\mathbb{S}^{(l)}$ to $Sig(l)$, CPS works for the hidden states $\boldsymbol{H}^{(l)}(\boldsymbol{x}) \in \mathbb{R}^{n \times d}$ of a TSFM layer during forward propagation as follows:

1. Centering $\boldsymbol{H}^{(l)}(\boldsymbol{x})$ by subtracting the mean across hidden state positions to obtain $\boldsymbol{H}_c^{(l)}(\boldsymbol{x}) = \boldsymbol{H}^{(l)}(\boldsymbol{x}) - \bar{\boldsymbol{h}}^{(l)}(\boldsymbol{x})$;

2. Computing the projections on signal subspaces $\Pi_{\mathbb{S}^{(l)}} \boldsymbol{H}_c^{(l)}(\boldsymbol{x})$ at all positions;

3. Scaling the signal components by a factor $\lambda$ so that $\tilde{\boldsymbol{H}}_c^{(l)}(\boldsymbol{x}) = \boldsymbol{H}_c^{(l)}(\boldsymbol{x}) + (\lambda - 1)\Pi_{\mathbb{S}^{(l)}} \boldsymbol{H}_c^{(l)}(\boldsymbol{x})$;

4. Adding back the mean to obtain the intervened hidden states $\tilde{\boldsymbol{H}}^{(l)}(\boldsymbol{x}) = \tilde{\boldsymbol{H}}_c^{(l)}(\boldsymbol{x}) + \bar{\boldsymbol{h}}^{(l)}(\boldsymbol{x})$.

The intervened hidden states are inputted to the next layer. We center the hidden states in Step 1 to emphasize the activation differences across positions. Arranging the basis of $\mathbb{S}^{(l)}$ into an orthogonal matrix $\boldsymbol{P} = [\boldsymbol{e}_1, \ldots, \boldsymbol{e}_k] \in \mathbb{R}^{k \times d}$, where $\boldsymbol{e}_i$ is the indicator vector of a signal neuron, the projection in Step 2 can be computed by matrix product. The CPS operation can be formulated simply as $\tilde{\boldsymbol{H}}^{(l)}(\boldsymbol{x}) = \boldsymbol{H}^{(l)}(\boldsymbol{x}) + (\lambda - 1)\boldsymbol{H}_c^{(l)}(\boldsymbol{x})\boldsymbol{P}^T\boldsymbol{P}$, which can be efficiently computed at each layer in $\mathcal{O}(ndk)$ cost, with $k \ll d$. The cost can be further reduced to $O(nk)$ leveraging sparsity of $\boldsymbol{P}$.

The CPS operation has desirable properties. First, the mean of hidden state neuron activations is unaltered by the operation, and the standard deviation scales proportionally with $\lambda$, which makes the operation easy to control and causes no distribution drift to neuron activations. Moreover, different from prior approaches that add a static steering vector to hidden representations [22, 25, 43], our approach adaptively alters neuron activations based on their distributions, improving the contrast of hidden states and clustering effects. We mathematically show that in many cases the CPS operation can reduce the cosine similarity between hidden states (see proofs in Appendix B).

We further propose an adaptive scaling approach that helps identify the cases necessary to apply the intervention. Since the signal activity scores $\mathcal{A}_{signal}^{(l)}$ measure neuron activity in the presence of

strong signals, we use them as a reference. At each layer, we compute the mean activity scores of signal neurons $\bar{\mathcal{A}}^{(l)}(\boldsymbol{x}) = \frac{1}{k} \sum_{j \in Sig(l)} \mathcal{A}^{(l)}(j \mid \boldsymbol{x})$. We then compute the adaptive scaling factor as a ratio $\lambda^{(l)} = \bar{\mathcal{A}}^{(l)}_{signal} / \bar{\mathcal{A}}^{(l)}(\boldsymbol{x})$, with $\bar{\mathcal{A}}^{(l)}_{signal} = \frac{1}{k} \sum_{j \in Sig(l)} \mathcal{A}^{(l)}_{signal}(j)$, and apply the intervention only when $\lambda^{(l)} > 1$. In this way, we selectively intervene layers with weak signal information and scale the activations of signal neurons to match those of the reference. We call the complete test-time intervention approach Signal Subspace Intervention through Magnification (SSIM).

# 5   Experiments

In this section, we conduct experiments to address the following questions: (1) How do hallucinations affect the performance of each type of TSFM? (2) What is the effect of our proposed test-time intervention on hallucination mitigation? (3) What is the performance of our proposed signal strength measures? (4) How does each our designed component impact the intervention performance?

## 5.1   Experimental Settings

**Datasets.** We curate a synthetic dataset comprising of common waveforms of sine, square, sawtooth, triangle, and pulse waves with varying slopes from $\{-0.01, 0, 0.01\}$. We vary the number of periods in the context from $\{8, 10, 12, 14, 16, 18, 20\}$ and the standard deviation of Gaussian noise added to the context signal from $\{0, 0.1, 0.2, 0.3, 0.4\}$. In addition, we adopt read-world datasets from the GIFT-Eval [3] benchmark covering various domains. We take a fixed number of final observations from each time series, dividing them into context and ground truth of fixed lengths. We discard time series instances with over $10\%$ missing values and impute missing values with the segment mean. As defined in §3, we retain time series instances whose ground truth satisfies the knowledge rules extracted from the context such that the context contains sufficient information for forecasting. Each dataset is randomly split into validation ($20\%$) and test ($80\%$) sets. Further details are provided in Appendix D.2.

**Baselines.** For hallucination mitigation, we compare SSIM with input denoising by smoothing as well as input perturbation with output averaging [25]. Additional comparison study with finetuning is presented in Appendix D.4. For hallucination detection, we compare SSAS with other statistics discussed in §4.1, including the mean pairwise cosine similarity of hidden states and the mean standard deviation of neuron activations.

**Evaluation metrics.** We evaluate forecast quality with $R^2$ and the Pearson correlation coefficient, which are scale invariant. Metric $R^2$ measures the goodness of fit of the forecast to the ground truth, ranging in $(-\infty, 1]$; the Pearson correlation coefficient measures the strength and direction of the linear relationship between the forecast and ground truth, ranging in $[-1, 1]$ (invalid values are imputed with 0). Whether a forecast is hallucinated is determined according to the knowledge rules defined in §3. We evaluate the effect of hallucination mitigation with hallucination rate reduction and forecast quality improvement. We assess the accuracy of hallucination detection with AUROC and forecasting performance prediction with the Spearman rank correlation coefficient.

**Implementation details.** We evaluate on three mainstream TSFMs: *Chronos* [4], *Chronos-Bolt*, and *TimesFM* [11]. We set the context length to 500 and the forecasting horizon to 64 for zero-shot time series forecasting in our main experiments, using the base versions of *Chronos* and *Chronos-Bolt* together with TimesFM-2.0. As *Chronos* produces probabilistic forecasts, we set the number of decoding samples to 1 and fix the random seed to ensure reproducibility. We set the frequency configuration of *TimesFM* to 0. For hallucination check, we set the tolerance thresholds $\delta$ of the trend, frequency, pattern, and ARMA rules to 0.25, 0.5, 0.5, and 0.25, respectively, based on validation. For SSIM, we perform grid search for the proportion of selected top neurons $\epsilon \in \{0.1, 0.2, 0.3, 0.4, 0.5\}$ and set it to 0.1 for Chronos and TimesFM and 0.2 for Chronos-Bolt based on validation. For baselines methods, we denoise input time series using the mean of sliding windows of size 5. We perturb input time series by Gaussian noise with a standard deviation of 0.05 times that of the input for 10 runs with different random seeds. We leverage signal strength measures for forecasting performance prediction and their negations for hallucination detection.

Table 1: Comparison of forecasting performance across domains, with the best results boldfaced.

| Model | Domain | Original | | | Denoising | | | Perturbation Averaging | | | SSIM (ours) | | |
|---|---|---|---|---|---|---|---|---|---|---|---|---|---|
| | | $Hal \downarrow$ | $R^2 \uparrow$ | $Corr \uparrow$ | $Hal \downarrow$ | $R^2 \uparrow$ | $Corr \uparrow$ | $Hal \downarrow$ | $R^2 \uparrow$ | $Corr \uparrow$ | $Hal \downarrow$ | $R^2 \uparrow$ | $Corr \uparrow$ |
| Chronos | Synthetic | 0.4524 | -0.1625 | 0.6265 | 0.4429 | -0.6734 | 0.5714 | 0.4333 | -0.1053 | 0.6392 | **0.4145** | **0.1854** | **0.7150** |
| | Econ/Fin | 0.4115 | -3.3554 | 0.4751 | 0.5007 | -4.5011 | 0.3413 | 0.4609 | -3.5727 | 0.4735 | **0.4061** | **-3.2037** | **0.5146** |
| | Energy | 0.1389 | -0.4839 | 0.7180 | 0.2504 | -3.0764 | 0.5315 | 0.1212 | -0.2073 | 0.7241 | **0.1191** | **0.0268** | **0.7707** |
| | Nature | 0.8035 | -10.7283 | 0.0457 | 0.9514 | -8.3558 | 0.0575 | 0.8436 | -7.2973 | 0.0552 | **0.6715** | **-0.7575** | **0.1082** |
| | Transport | 0.4197 | -1.6444 | 0.5127 | 0.7565 | -1.4804 | 0.3295 | 0.4461 | -1.4582 | 0.5315 | **0.3938** | **-0.2221** | **0.6081** |
| | WebOps | **0.5801** | -414.8937 | 0.2762 | 0.8833 | -139.8035 | 0.1559 | 0.6115 | -79.3298 | 0.2822 | 0.6052 | **-21.8389** | **0.3369** |
| | Aggregated | 0.4531 | -82.3762 | 0.4458 | 0.5991 | -28.6399 | 0.3336 | 0.4759 | -17.2700 | 0.4529 | **0.4231** | **-5.0845** | **0.5061** |
| Chronos-Bolt | Synthetic | 0.5381 | 0.0152 | 0.5589 | 0.5810 | -0.0625 | 0.5302 | 0.5500 | 0.0099 | 0.5586 | **0.5231** | **0.0238** | **0.5672** |
| | Econ/Fin | 0.4856 | -1.3759 | 0.5727 | 0.4870 | **-1.2344** | 0.4243 | 0.4911 | -1.4191 | **0.5929** | **0.4787** | -1.2891 | 0.5811 |
| | Energy | 0.0985 | 0.1499 | 0.7694 | 0.1712 | -0.0411 | 0.6291 | 0.1002 | 0.1033 | 0.7671 | **0.0843** | **0.1508** | **0.7765** |
| | Nature | 0.9426 | -0.0744 | 0.1400 | 0.9536 | -0.6290 | 0.1057 | 0.9404 | -0.0876 | 0.1462 | **0.9316** | -0.0657 | **0.1472** |
| | Transport | 0.6684 | 0.2039 | 0.6501 | 0.8446 | -0.2129 | 0.4072 | 0.6632 | 0.2015 | 0.6488 | **0.6522** | **0.2124** | **0.6563** |
| | WebOps | 0.6777 | -0.6529 | 0.3591 | 0.9024 | -1.2424 | 0.1777 | 0.6707 | **-0.4822** | 0.3625 | **0.6632** | -0.6680 | **0.3646** |
| | Aggregated | 0.5308 | -0.4260 | 0.5099 | 0.6084 | -0.6662 | 0.3848 | 0.5321 | -0.4163 | 0.5158 | **0.5191** | -0.3766 | **0.5171** |
| TimesFM | Synthetic | 0.1143 | 0.5661 | 0.9143 | 0.1452 | 0.4685 | 0.7568 | 0.1190 | 0.5688 | 0.9094 | **0.1049** | **0.5699** | **0.9194** |
| | Econ/Fin | 0.3868 | -1.8715 | 0.7793 | 0.4472 | -5.1532 | 0.3722 | 0.4005 | -1.8277 | 0.7657 | **0.3771** | **-0.3161** | **0.7847** |
| | Energy | 0.1357 | 0.2745 | 0.8065 | 0.1987 | -0.0981 | 0.6150 | 0.1341 | **0.2916** | 0.7941 | **0.1222** | 0.1304 | **0.8133** |
| | Nature | 0.9558 | -0.1678 | **0.1620** | 0.9691 | -0.2561 | 0.1302 | **0.9492** | -0.1715 | 0.1451 | 0.9536 | -0.0902 | 0.1559 |
| | Transport | 0.5751 | **0.4245** | 0.7027 | 0.6321 | -0.4210 | 0.3540 | **0.5648** | 0.4236 | 0.7028 | 0.5733 | 0.4201 | 0.7076 |
| | WebOps | 0.6429 | -22.0090 | 0.4224 | 0.8676 | -20.0077 | 0.2324 | **0.6202** | -7.5448 | 0.4216 | 0.6359 | **-6.7480** | 0.4291 |
| | Aggregated | 0.4441 | -4.5461 | 0.6368 | 0.5251 | -5.3271 | 0.4119 | 0.4418 | -2.0435 | 0.6275 | **0.4348** | **-1.2529** | **0.6410** |

## 5.2 Main Experimental Results

**TSFM hallucinations (*RQ1*).** Table 1 summarizes the performance of original TSFMs. We note that the hallucination rate varies drastically across domains. On domain *Energy* the time series have more periodic patterns, resulting in low hallucination rates; while on domain *Nature* the time series contain more abrupt changes, making it harder for TSFMs to process the context information. The forecasts appear to have stronger correlations with the ground truths on domains where the hallucination rate is lower. Table 2 compares the performance of hallucinated forecasts versus non-hallucinated forecasts by TSFMs. We observe that hallucinated forecasts are consistently outperformed by non-hallucinated forecasts. For Chronos-Bolt and TimesFM, the mean $R^2$ of non-hallucinated forecasts stays positive on each domain. Hallucinated forecasts have substantially weaker correlations with ground truths than non-hallucinated forecasts for all models, indicating weaker signal information captured from the context. Unpaired $t$-tests yield $p < 10^{-5}$ against the null hypothesis that the performance of hallucinated and non-hallucinated forecasts is the same. These results underscore the significance of TSFM forecasting performance being affected by hallucinations.

Figure 4 (a) compares the distributions of TSFM hallucinations, with Type 1 referring to the violation of the trend rule, Type 2 to the frequency rule, and Type 3 to both the pattern and ARMA rules. We observe that Type 3 hallucinations are the most common ones, since the pattern and ARMA rules demand more sophisticated reasoning of context information. Chronos exhibits fewer Type 1 and Type 2 hallucinations than the other models as it does not apply input patching, which enables more accurate processing of trend and frequency information.

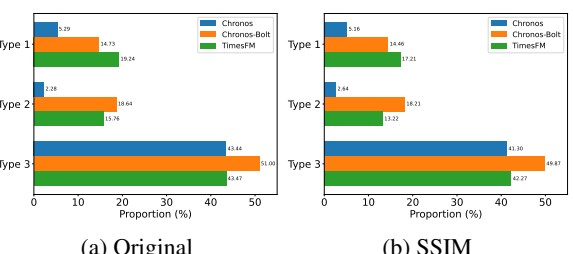

(a) Original    (b) SSIM

Figure 4: Distributions of hallucination types.

**Hallucination mitigation (*RQ2*).** Table 1 compares the forecasting performance with SSIM and the baseline methods. SSIM attains the best performance overall, yielding up to $6.62\%$ reduction on hallucination rate, $93.83\%$ gain on $R^2$, and $13.52\%$ gain on correlation over the original models. Although denoising improves $R^2$ in some cases by reducing the impact of outliers, it causes higher hallucination rates and lower correlation in general due to the loss of context information. Perturbation averaging improves the forecast quality to some extent, but it does not sufficiently mitigate hallucinations and demands considerably more computation. In comparison, SSIM pre-computes the signal neurons only once for each TSFM and incurs minor additional overheads during test time. The performance margin between SSIM and baselines is statistically significant at $p < 0.01$ by the Friedman-Nemenyi test [15, 29]. We further analyze the effect of SSIM on the distributions of hallucinations. From Figure 4 (b), we note that SSIM has the greatest effect on Type 3 hallucinations,

Table 2: Performance comparison of hallucinated and non-hallucinated forecasts by TSFMs.

| Metric | Domain | Chronos | | | Chronos-Bolt | | | TimesFM | | |
|---|---|---|---|---|---|---|---|---|---|---|
| | | *Hal* | *Non-hal* | *Diff* | *Hal* | *Non-hal* | *Diff* | *Hal* | *Non-hal* | *Diff* |
| $R^2 \uparrow$ | Synthetic | -1.1206 | 0.6290 | 1.7496 | -0.4450 | 0.5512 | 0.9962 | -1.5404 | 0.8379 | 2.3783 |
| | Econ/Fin | -6.7964 | -0.9492 | 5.8473 | -3.3098 | 0.4497 | 3.7595 | -5.7181 | 0.5553 | 6.2734 |
| | Energy | -2.8794 | -0.0974 | 2.7820 | -1.1625 | 0.2934 | 1.4559 | -0.8971 | 0.4585 | 1.3556 |
| | Nature | -12.8573 | -2.0209 | 10.8364 | -0.0846 | 0.0933 | 0.1779 | -0.1923 | 0.3626 | 0.5549 |
| | Transport | -3.9208 | 0.0018 | 3.9226 | -0.0548 | 0.7253 | 0.7800 | 0.2046 | 0.7223 | 0.5178 |
| | WebOps | -634.8971 | -110.9056 | 523.9916 | -1.1848 | 0.4657 | 1.6505 | -34.4655 | 0.4127 | 34.8783 |
| | Aggregated | -161.6823 | -16.6599 | 145.0224 | -1.1647 | 0.4096 | 1.5743 | -10.9572 | 0.5757 | 11.5329 |
| $Corr \uparrow$ | Synthetic | 0.3478 | 0.8568 | 0.5090 | 0.3596 | 0.7910 | 0.4314 | 0.7436 | 0.9364 | 0.1927 |
| | Econ/Fin | 0.0853 | 0.7476 | 0.6624 | 0.2558 | 0.8719 | 0.6162 | 0.5927 | 0.8969 | 0.3042 |
| | Energy | 0.4738 | 0.7574 | 0.2836 | 0.5716 | 0.7910 | 0.2194 | 0.6300 | 0.8342 | 0.2043 |
| | Nature | 0.0226 | 0.1401 | 0.1176 | 0.0991 | 0.8118 | 0.7127 | 0.1312 | 0.8289 | 0.6977 |
| | Transport | 0.2776 | 0.6827 | 0.4051 | 0.5407 | 0.8708 | 0.3301 | 0.5744 | 0.8765 | 0.3021 |
| | WebOps | 0.1036 | 0.5148 | 0.4113 | 0.1760 | 0.7441 | 0.5681 | 0.2135 | 0.7984 | 0.5849 |
| | Aggregated | 0.1459 | 0.6943 | 0.5484 | 0.2441 | 0.8105 | 0.5664 | 0.3429 | 0.8716 | 0.5286 |

Table 3: The results of TSFM hallucination detection and forecasting performance prediction, with the best results boldfaced. For each method, the first column shows AUROC of hallucination detection and the latter two columns show rank correlations with the performance metrics. The statistical significance of positive rank correlations is indicated with * for $p < 0.05$ and ** for $p < 0.01$.

| Model | Domain | Cosine Similarity | | | Activation Variance | | | SSAS (Ours) | | |
|---|---|---|---|---|---|---|---|---|---|---|
| | | $Hal \uparrow$ | $R^2 \uparrow$ | $Corr \uparrow$ | $Hal \uparrow$ | $R^2 \uparrow$ | $Corr \uparrow$ | $Hal \uparrow$ | $R^2 \uparrow$ | $Corr \uparrow$ |
| Chronos | Synthetic | 0.7847 | 0.3834** | 0.3262** | 0.6786 | 0.3734** | 0.4052** | **0.8316** | **0.4299**** | **0.5111**** |
| | Econ/Fin | **0.8495** | **0.6501**** | **0.6208**** | 0.6927 | 0.5096** | 0.5507** | 0.7833 | 0.5034** | 0.5258** |
| | Energy | 0.7124 | **0.5088**** | **0.3528**** | 0.8093 | -0.1116 | 0.0373 | **0.8096** | 0.1166** | 0.0363 |
| | Nature | 0.4978 | 0.3382** | 0.1524** | 0.5384 | **0.3490**** | **0.1886**** | **0.5925** | 0.3430** | 0.1507** |
| | Transport | 0.6601 | 0.4706** | 0.5550** | **0.7466** | **0.5351**** | **0.6083**** | 0.6767 | 0.4158** | 0.5234** |
| | WebOps | 0.5542 | **0.2328**** | **0.3710**** | 0.5060 | 0.1363** | 0.2720** | **0.5740** | 0.1693** | 0.2526** |
| | Aggregated | 0.7903 | **0.5866**** | **0.5804**** | 0.7226 | 0.4197** | 0.5277** | **0.8086** | 0.5082** | 0.5758** |
| Chronos-Bolt | Synthetic | 0.4416 | -0.1767 | -0.0361 | 0.4011 | -0.3806 | -0.3569 | **0.5282** | **0.2363**** | **0.2142**** |
| | Econ/Fin | 0.2247 | -0.5253 | -0.5530 | 0.3851 | -0.3600 | -0.4283 | **0.8528** | **0.6289**** | **0.5979**** |
| | Energy | 0.5360 | 0.2052** | 0.1488** | 0.4739 | **0.4158**** | **0.3741**** | **0.7451** | -0.0603 | -0.0865 |
| | Nature | 0.9343 | -0.1155 | 0.3819** | **0.9395** | -0.1003 | **0.3842**** | 0.6340 | **0.0895*** | 0.1690** |
| | Transport | **0.7183** | 0.2601** | 0.2394** | 0.6656 | 0.2730** | 0.2064** | 0.6416 | **0.2850**** | **0.2858**** |
| | WebOps | **0.6656** | 0.1315** | **0.4091**** | 0.5224 | -0.0236 | 0.1767** | 0.6518 | **0.2188**** | 0.3423** |
| | Aggregated | 0.6279 | 0.0462* | 0.2607** | 0.6131 | 0.0789** | 0.1840** | **0.7991** | **0.3860**** | **0.5037**** |
| TimesFM | Synthetic | 0.4892 | -0.2210 | -0.1684 | 0.3821 | -0.3302 | -0.2738 | **0.4902** | **-0.0081** | **-0.0359** |
| | Econ/Fin | 0.2210 | -0.6896 | -0.6568 | 0.1930 | -0.5539 | -0.5192 | **0.4625** | **-0.0932** | **-0.0987** |
| | Energy | 0.2898 | -0.2597 | -0.1570 | 0.2777 | -0.1531 | -0.1911 | **0.7469** | **0.1407**** | **0.1042*** |
| | Nature | 0.4042 | -0.0565 | -0.0041 | 0.7912 | 0.0587 | 0.3480** | **0.8934** | **0.1744**** | **0.3870**** |
| | Transport | 0.4881 | -0.0669 | -0.0174 | 0.6121 | 0.1892** | 0.2710** | **0.6529** | **0.3483**** | **0.3622**** |
| | WebOps | 0.3756 | -0.3497 | -0.2117 | 0.5615 | -0.1001 | 0.1046* | **0.6365** | **0.0637** | **0.2418**** |
| | Aggregated | 0.3963 | -0.2608 | -0.1767 | 0.5211 | -0.0517 | 0.0634* | **0.6890** | **0.2636**** | **0.3552**** |

yielding up to $5\%$ reduction. The facilitation of signal information propagation helps a TSFM better capture patterns from the context.

**Hallucination detection and performance prediction (*RQ3*).** We report the performance of different measures for hallucination detection and forecasting performance prediction in Table 3. SSAS has consistently strong predictive power across domains for different TSFMs, attaining high AUROC for hallucination detection and significantly positive rank correlations with forecasting performance, demonstrating the effectiveness of our proposed signal strength measure and the critical role of signal neurons in generating high-quality forecasts. Simply using the mean neuron activation variance as a measure yields inferior and less consistent results overall, as it is obscured by the activity of irrelevant neurons. The mean pairwise cosine similarity of hidden states exhibits relatively strong predictive power for Chronos and Chronos-Bolt but fails to generalize to TimesFM.

**Ablation study (*RQ4*).** We compare the performance of SSIM intervention with the following reduced variants: (1) w/o adaptive scaling: using a constant scaling factor $\lambda$ for each layer; (2) w/o centering: scaling neuron activations without subtracting the mean across the positions of a layer [10, 40]. From Figure 5, SSIM consistently outperforms the variants. The performance

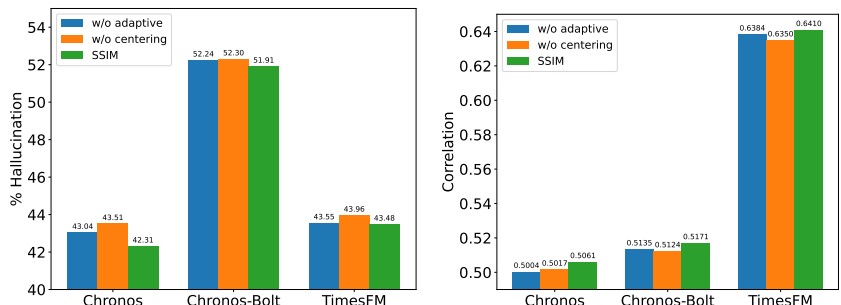

Figure 5: The aggregated mean performance of SSIM and the variants for TSFMs.

differences are significant at $p < 0.01$ by paired $t$-tests, highlighting the effectiveness of our design. The adaptive scaling enables more refined control of the intervention, providing greater magnification when weak signal information is detected at a layer. The centering operation places an emphasis on activation differences that helps reduce the homogeneity of hidden states without causing distribution drifts to the activations of intervened neurons.

# 6    Conclusion

Time series foundation models represent a promising paradigm for time series analyses and forecasting, yet the issue of hallucinations has been underexplored in existing literature. In this paper, we have formally defined TSFM hallucinations in the zero-shot forecasting setting and outlined a set of knowledge rules for checking hallucinations in practice. We have found that hallucinations are associated with a lack of context information in hidden states through experimental analyses. We have proposed a method to identify the signal spaces along with a measure to quantify the signal strength of hidden states. We have further developed an intervention approach that mitigates hallucinations by magnifying the signal information of hidden states. Extensive experiments across various domains have demonstrated that our test-time intervention effectively mitigates hallucinations and improves the quality of TSFM forecasts. The signal strength measure we proposed has shown strong predictive power of both hallucinations and forecasting performance. Our work contributes to deeper understanding of TSFM trustworthiness that could foster future research.

# Acknowledgments

Yufeng Zou is partially supported by the Walter P. Murphy Fellowship. The content is solely the responsibility of the authors and does not necessarily represent the official views of the funding agencies.

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

# Appendices

## A  Knowledge Rules

The rolling windows we apply on the context always have the same size as the forecasting horizon. We adopt the OLS and ARMA [6] implementations in the `statsmodels` package[3,4]. We adopt the STFT [18] implementation in the `SciPy` package[5], with unsymmetrical Parzen windows and hop=1.

## B  Properties of CPS Operation

Center-Project-Scale (CPS) is our proposed intervention operation that magnifies the signal information in hidden states with a scaling factor $\lambda$. We use $\mathbb{S}$ to denote the identified signal subspaces. CPS has several mathematical properties that help with the effectiveness of hidden state intervention:

**Theorem 1** (**Mean Invariance**). *The mean of hidden state neuron activations across the positions of a layer is unaltered by the CPS operation, i.e., $\mathbb{E}(\tilde{\boldsymbol{h}}) = \bar{\boldsymbol{h}}$.*

*Proof.* Let $\bar{\boldsymbol{h}}$ denote the mean of hidden states across the positions of a layer. For a hidden state $\boldsymbol{h}$, the hidden state after CPS becomes $\tilde{\boldsymbol{h}} = \boldsymbol{h} + (\lambda - 1)\Pi_{\mathbb{S}}(\boldsymbol{h} - \bar{\boldsymbol{h}})$. The mean of hidden states after CPS is $\mathbb{E}(\tilde{\boldsymbol{h}}) = \frac{1}{n}\sum_{i=1}^{n}\tilde{\boldsymbol{h}}_i = \frac{1}{n}\sum_{i=1}^{n}[\boldsymbol{h}_i + (\lambda-1)\Pi_{\mathbb{S}}(\boldsymbol{h}_i - \bar{\boldsymbol{h}})] = \frac{1}{n}\sum_{i=1}^{n}\boldsymbol{h}_i + \frac{1}{n}\sum_{i=1}^{n}(\lambda-1)\Pi_{\mathbb{S}}(\boldsymbol{h}_i - \bar{\boldsymbol{h}})$. Since $\frac{1}{n}\sum_{i=1}^{n}(\lambda - 1)\Pi_{\mathbb{S}}(\boldsymbol{h}_i - \bar{\boldsymbol{h}}) = (\lambda - 1)\Pi_{\mathbb{S}}[\frac{1}{n}\sum_{i=1}^{n}(\boldsymbol{h}_i - \bar{\boldsymbol{h}})] = (\lambda - 1)\Pi_{\mathbb{S}}(\bar{\boldsymbol{h}} - \bar{\boldsymbol{h}}) = \boldsymbol{0}$, we have $\mathbb{E}(\tilde{\boldsymbol{h}}) = \frac{1}{n}\sum_{i=1}^{n}\boldsymbol{h}_i + \boldsymbol{0} = \bar{\boldsymbol{h}}$. $\qquad\square$

**Theorem 2** (**Standard Deviation Scaling**). *In CPS, the standard deviation of signal neuron activations across the positions of a layer scales with $\lambda$, i.e., $\tilde{\mathcal{A}}(j) = \lambda\,\mathcal{A}(j)$ for any $j \in \mathbb{S}$.*

---

[3]`https://www.statsmodels.org/stable/generated/statsmodels.regression.linear_model.OLS.html`

[4]`https://www.statsmodels.org/stable/generated/statsmodels.tsa.arima.model.ARIMA.html`

[5]`https://docs.scipy.org/doc/scipy/reference/generated/scipy.signal.ShortTimeFFT.html#scipy.signal.ShortTimeFFT`

*Proof.* For any $j \in \mathbb{S}$,

$$
\begin{aligned}
\tilde{\mathcal{A}}(j) &= \sqrt{\frac{1}{n}\sum_{i=1}^{n}\left(\tilde{\boldsymbol{H}}_{i,j} - \mathbb{E}(\tilde{\boldsymbol{h}}_j)\right)^2} \\
&= \sqrt{\frac{1}{n}\sum_{i=1}^{n}\left[\boldsymbol{H}_{i,j} + (\lambda-1)\Pi_{\mathbb{S}}(\boldsymbol{H}_{i,j} - \bar{\boldsymbol{h}}_j) - \bar{\boldsymbol{h}}_j\right]^2} \qquad \text{(From Theorem 1)} \\
&= \sqrt{\frac{1}{n}\sum_{i=1}^{n}\left[\boldsymbol{H}_{i,j} + (\lambda-1)(\boldsymbol{H}_{i,j} - \bar{\boldsymbol{h}}_j) - \bar{\boldsymbol{h}}_j\right]^2} \\
&= \sqrt{\frac{1}{n}\sum_{i=1}^{n}\left(\lambda\boldsymbol{H}_{i,j} - \lambda\bar{\boldsymbol{h}}_j\right)^2} \\
&= \lambda\sqrt{\frac{1}{n}\sum_{i=1}^{n}\left(\boldsymbol{H}_{i,j} - \bar{\boldsymbol{h}}_j\right)^2} \\
&= \lambda\,\mathcal{A}(j)\,.
\end{aligned}
$$

$\square$

**Theorem 3** (Cosine Similarity Reduction). *For two hidden states $\boldsymbol{h}_1$ and $\boldsymbol{h}_2$ that have different projection directions in the signal subspace after subtracting the mean, the CPS operation reduces the cosine similarity of these two hidden states with a sufficiently large scaling factor $\lambda$, i.e., $cos(\tilde{\boldsymbol{h}}_1, \tilde{\boldsymbol{h}}_2) < cos(\boldsymbol{h}_1, \boldsymbol{h}_2)$ when $\lambda > \max\{-\frac{(\boldsymbol{s}_1+\boldsymbol{s}_2)\cdot\bar{\boldsymbol{h}}}{\boldsymbol{s}_1\cdot\boldsymbol{s}_2} - 1, -\frac{2\boldsymbol{s}_1\cdot\bar{\boldsymbol{h}}}{\|\boldsymbol{s}_1\|^2} - 1, -\frac{2\boldsymbol{s}_2\cdot\bar{\boldsymbol{h}}}{\|\boldsymbol{s}_2\|^2} - 1, 1\}$.*

*Proof.* The original hidden states can be decomposed as $\boldsymbol{h}_1 = \boldsymbol{s}_1 + \boldsymbol{n}_1 + \bar{\boldsymbol{h}}$ and $\boldsymbol{h}_2 = \boldsymbol{s}_2 + \boldsymbol{n}_2 + \bar{\boldsymbol{h}}$, where $\boldsymbol{s}_1 = \Pi_{\mathbb{S}}(\boldsymbol{h}_1 - \bar{\boldsymbol{h}})$, $\boldsymbol{n}_1 = \Pi_{\bar{\mathbb{S}}}(\boldsymbol{h}_1 - \bar{\boldsymbol{h}})$, $\boldsymbol{s}_2 = \Pi_{\mathbb{S}}(\boldsymbol{h}_2 - \bar{\boldsymbol{h}})$, and $\boldsymbol{n}_2 = \Pi_{\bar{\mathbb{S}}}(\boldsymbol{h}_2 - \bar{\boldsymbol{h}})$. We have $\boldsymbol{h}_1 \cdot \boldsymbol{h} = \boldsymbol{s}_1 \cdot \boldsymbol{s}_2 + (\boldsymbol{s}_1 + \boldsymbol{s}_2) \cdot \bar{\boldsymbol{h}} + \text{residual terms}$. The hidden states after CPS can be written as $\tilde{\boldsymbol{h}}_1 = \lambda\boldsymbol{s}_1 + \boldsymbol{n}_1 + \bar{\boldsymbol{h}}$ and $\tilde{\boldsymbol{h}}_2 = \lambda\boldsymbol{s}_2 + \boldsymbol{n}_2 + \bar{\boldsymbol{h}}$. We have $\tilde{\boldsymbol{h}}_1 \cdot \tilde{\boldsymbol{h}}_2 = \lambda^2\boldsymbol{s}_1 \cdot \boldsymbol{s}_2 + \lambda(\boldsymbol{s}_1 + \boldsymbol{s}_2) \cdot \bar{\boldsymbol{h}} + \text{residual terms}$. The residual terms are $\boldsymbol{n}_1 \cdot \boldsymbol{n}_2 + (\boldsymbol{n}_1 + \boldsymbol{n}_2) \cdot \bar{\boldsymbol{h}} + \|\bar{\boldsymbol{h}}\|^2$ in both cases. Then,

$$
\begin{aligned}
& \tilde{\boldsymbol{h}}_1 \cdot \tilde{\boldsymbol{h}}_2 < \boldsymbol{h}_1 \cdot \boldsymbol{h}_2 \\
\Longleftrightarrow \quad & \lambda^2\boldsymbol{s}_1 \cdot \boldsymbol{s}_2 + \lambda(\boldsymbol{s}_1 + \boldsymbol{s}_2) \cdot \bar{\boldsymbol{h}} < \boldsymbol{s}_1 \cdot \boldsymbol{s}_2 + (\boldsymbol{s}_1 + \boldsymbol{s}_2) \cdot \bar{\boldsymbol{h}} \\
\Longleftrightarrow \quad & (\lambda^2 - 1)\boldsymbol{s}_1 \cdot \boldsymbol{s}_2 + (\lambda - 1)(\boldsymbol{s}_1 + \boldsymbol{s}_2) \cdot \bar{\boldsymbol{h}} < 0 \\
\Longleftrightarrow \quad & (\lambda + 1)\boldsymbol{s}_1 \cdot \boldsymbol{s}_2 + (\boldsymbol{s}_1 + \boldsymbol{s}_2) \cdot \bar{\boldsymbol{h}} < 0 \qquad \text{(Assume } \lambda > 1) \\
\Longleftrightarrow \quad & \lambda > -\frac{(\boldsymbol{s}_1 + \boldsymbol{s}_2) \cdot \bar{\boldsymbol{h}}}{\boldsymbol{s}_1 \cdot \boldsymbol{s}_2} - 1\,. \qquad \text{(Assume } \boldsymbol{s}_1 \cdot \boldsymbol{s}_2 < 0)
\end{aligned}
$$

Also,

$$
\begin{aligned}
& \|\tilde{\boldsymbol{h}}_1\| > \|\boldsymbol{h}_1\| \\
\Longleftrightarrow \quad & \lambda^2\boldsymbol{s}_1 \cdot \boldsymbol{s}_1 + 2\lambda\boldsymbol{s}_1 \cdot \bar{\boldsymbol{h}} + \|\bar{\boldsymbol{h}}\|^2 > \boldsymbol{s}_1 \cdot \boldsymbol{s}_1 + 2\boldsymbol{s}_1 \cdot \bar{\boldsymbol{h}} + \|\bar{\boldsymbol{h}}\|^2 \\
\Longleftrightarrow \quad & (\lambda^2 - 1)\boldsymbol{s}_1 \cdot \boldsymbol{s}_1 + 2(\lambda - 1)\boldsymbol{s}_1 \cdot \bar{\boldsymbol{h}} > 0 \\
\Longleftrightarrow \quad & (\lambda + 1)\boldsymbol{s}_1 \cdot \boldsymbol{s}_1 + 2\boldsymbol{s}_1 \cdot \bar{\boldsymbol{h}} > 0 \qquad \text{(Assume } \lambda > 1) \\
\Longleftrightarrow \quad & \lambda > -\frac{2\boldsymbol{s}_1 \cdot \bar{\boldsymbol{h}}}{\|\boldsymbol{s}_1\|^2} - 1\,.
\end{aligned}
$$

Similarly, we have $\|\tilde{\boldsymbol{h}}_2\| > \|\boldsymbol{h}_2\|$ when $\lambda > -\frac{2\boldsymbol{s}_2\cdot\bar{\boldsymbol{h}}}{\|\boldsymbol{s}_2\|^2} - 1$ and $\lambda > 1$.

Hence, when $\lambda > \max\{-\frac{(\boldsymbol{s}_1+\boldsymbol{s}_2)\cdot\bar{\boldsymbol{h}}}{\boldsymbol{s}_1\cdot\boldsymbol{s}_2} - 1, -\frac{2\boldsymbol{s}_1\cdot\bar{\boldsymbol{h}}}{\|\boldsymbol{s}_1\|^2} - 1, -\frac{2\boldsymbol{s}_2\cdot\bar{\boldsymbol{h}}}{\|\boldsymbol{s}_2\|^2} - 1, 1\}$ and $\boldsymbol{s}_1 \cdot \boldsymbol{s}_2 < 0$, we have

$$
cos(\tilde{\boldsymbol{h}}_1, \tilde{\boldsymbol{h}}_2) = \frac{\tilde{\boldsymbol{h}}_1 \cdot \tilde{\boldsymbol{h}}_2}{\|\tilde{\boldsymbol{h}}_1\|\|\tilde{\boldsymbol{h}}_2\|} < \frac{\boldsymbol{h}_1 \cdot \boldsymbol{h}_2}{\|\boldsymbol{h}_1\|\|\boldsymbol{h}_2\|} = cos(\boldsymbol{h}_1, \boldsymbol{h}_2)\,.
$$

Since we select signal neurons with top activity scores as the bases of signal subspaces, we often have $\boldsymbol{s}_1 \cdot \boldsymbol{s}_2 < 0$ when the time series information encoded in $\boldsymbol{h}_1$ and $\boldsymbol{h}_2$ is distinct. The CPS operation reduces hidden state homogeneity by increasing the contrast of time series signal information across the positions of a layer.

**Corollary 4.** *In CPS, the cosine similarity of two hidden states tends to the cosine similarity of their projections the signal subspace as the scaling factor tends to infinity, i.e.,* $\lim_{\lambda \to \infty} cos(\tilde{\boldsymbol{h}}_1, \tilde{\boldsymbol{h}}_2) = cos(\boldsymbol{s}_1, \boldsymbol{s}_2)$.

The proofs above explain how the CPS operation improves the clustering effects of hidden states by magnifying the time series signal information.

## C  SSIM Algorithm

Algorithm 1 details the full procedures of SSIM. The additional computation overhead at each layer is $O(nk)$, with $k$ being the number of signal neurons at the layer.

---

**Algorithm 1:** SSIM: Signal Subspace Intervention through Magnification

---

**Input** :TSFM $\mathcal{M}_\theta$ with $L$ layers, context time series $\boldsymbol{x}_{context}$, signal neurons $Sig$, reference
         neuron activity scores $\mathcal{A}_{signal}$
**Output :**Forecasts $\hat{\boldsymbol{x}}$

1   $\boldsymbol{H}^{(0)} \leftarrow \text{Preprocess}(\boldsymbol{x}_{context})$
2   **for** $l \leftarrow 1, \ldots, L$ **do**
3      $\boldsymbol{H}^{(l)} \leftarrow \mathcal{M}_\theta^{(l)}(\boldsymbol{H}^{(l-1)})$
4      $\bar{\mathcal{A}}^{(l)}(\boldsymbol{x}) \leftarrow \frac{1}{k} \sum_{j \in Sig(l)} \mathcal{A}^{(l)}(j \mid \boldsymbol{x})$
5      $\bar{\mathcal{A}}_{signal}^{(l)} \leftarrow \frac{1}{k} \sum_{j \in Sig(l)} \mathcal{A}_{signal}^{(l)}(j)$
6      $\lambda^{(l)} \leftarrow \frac{\bar{\mathcal{A}}_{signal}^{(l)}}{\bar{\mathcal{A}}^{(l)}(\boldsymbol{x})}$
7      **if** $\lambda^{(l)} > 1$ **then**
8          $\boldsymbol{H}_c^{(l)} \leftarrow \boldsymbol{H}^{(l)} - \bar{\boldsymbol{h}}^{(l)}$                    ▷ Center
9          $\boldsymbol{H}^{(l)} \leftarrow \boldsymbol{H}^{(l)} + (\lambda^{(l)} - 1)\boldsymbol{H}_c^{(l)}[:, Sig(l)]$     ▷ Project and scale
10     **end**
11 **end**
12 $\hat{\boldsymbol{x}} \leftarrow \text{Predict}(\boldsymbol{H}^{(L)})$
13 **return** $\hat{\boldsymbol{x}}$

---

## D  Experimental Details

### D.1  Experimental Platform

All experiments are conducted on the Ubuntu 22.04.4 LTS operating system, 16 Intel(R) Core(TM) i7-7820X CPUs, and 4 NVIDIA GeForce RTX 2080 Ti GPUs, with the framework of Python 3.11.9 and PyTorch 1.12.1.

### D.2  Dataset Details

**Synthetic dataset.** The times series we generate takes the form of $x(t) = signal(t) + trend(t) + noise(t)$. For signal component, we adopt common waveforms of sine, square, sawtooth, triangle, and pulse waves as implemented in the `SciPy` package[6]. We vary the number of signal periods in the context input from $\{8, 10, 12, 14, 16, 18, 20\}$. We vary the slope of the trend component from $\{-0.01, 0, 0.01\}$. The noise component is Gaussian noise with mean of 0, and we vary the standard deviation from $\{0, 0.1, 0.2, 0.3, 0.4\}$. In this way, we generate 525 time series instances in total.

---

[6]`https://docs.scipy.org/doc/scipy/reference/signal.html`

**Real-world dataset.** We adopt the GIFT-Eval benchmark [3]. We discard time series instances with over $10\%$ missing values and impute missing values with the mean of the segment. As explained in §3 of the main text, we retain time series instances whose ground truth satisfies the knowledge rules extracted from the context such that the context contains sufficient information for forecasting. The datasets are categorized into five different domains, with the bracket showing the number of time series instances after preprocessing:

- Econ/Fin: m4_daily (294), m4_hourly (391), m4_monthly (168), m4_quarterly (13), m4_weekly (42);

- Energy: electricity/15T (235), electricity/D (43), electricity/H (318), solar/10T (43), solar/H (132);

- Nature: kdd_cup_2018_with_missing/H (66), temperature_rain_with_missing (2358);

- Transport: LOOP_SEATTLE/5T (37), LOOP_SEATTLE/H (129), M_DENSE/D (9), M_DENSE/H (25), SZ_TAXI/15T (26), SZ_TAXI/H (12);

- WebOps: bitbrains_fast_storage/5T (261), bitbrains_fast_storage/H (203), bitbrains_rnd/5T (124), bitbrains_rnd/H (127).

To preserve data balance, we randomly sample $500$ time series instances from oversized datasets.

### D.3 Rule Ablations

In Table 4, we report the ablation results on the effect of each knowledge rule that constitutes our definition of time series forecasting hallucinations in §3 of the main text. We note that all three rules contribute to the differentiation of the forecast quality, with the aggregated Pearson correlation of non-hallucinated forecasts substantially higher than that of hallucinated forecasts in all cases. The pattern+ARMA rule effectively differentiates both $R^2$ and correlations in all cases. The frequency rule differentiates correlations in all cases but not $R^2$, affected by the negative $R^2$ values from the misalignment between forecasts and ground truths on some outlier test instances. The synergy of these rules gives the best performance differentiation overall.

We further examine the effect of pattern and ARMA rules separately. From Table 4, both pattern and ARMA rules contribute to the differentiation of TSFM forecast quality on their own, with positive differences of both $R^2$ and correlations in all cases. The pattern+ARMA rule narrows down the scope of hallucinations by considering forecasts that violate both the pattern and ARMA rules as hallucinations. It captures a smaller set of forecasting hallucinations with poor quality and thus provides superior performance differentiation overall.

### D.4 Additional Baseline

As a stronger hallucination mitigation baseline, we full-parameter finetune TSFMs using the validation data of each domain with 1,000 training steps. Table 5 compares the forecasting performance across domains. We observe that the performance of SSIM for Chronos is on par with full-parameter finetuning on most domains and better on some domains such as Energy, while finetuning suffers poor performance on some test instances due to overfitting. For TimesFM, finetuning suffers from even more severe overfitting, leading to inferior performance on all domains except Energy. Energy is a domain with frequent and regular patterns. SSIM saves the overheads of finetuning and better preserves the generalization performance of TSFMs.

### D.5 Parameter Study

Figure 6 reports the validation performance selecting varying proportions of top neurons for SSIM. We observe that while intervening a reasonable number of signal neurons boosts the forecasting performance of TSFMs, selecting too many neurons degrades the performance because the activations of noise neurons may also get scaled. Based on the results, we set this parameter to $0.1$ for Chronos and TimesFM and $0.2$ for Chronos-Bolt.

Table 4: Comparison of the aggregated mean performance between hallucinated and non-hallucinated TSFM forecasts by checking with different knowledge rules.

| Models | Rules | $R^2 \uparrow$ | | | $Corr \uparrow$ | | |
|---|---|---|---|---|---|---|---|
| | | *Hal* | *Non-hal* | *Diff* | *Hal* | *Non-hal* | *Diff* |
| Chronos | All | -161.6823 | -16.6599 | 145.0224 | 0.1459 | 0.6943 | 0.5484 |
| | Trend | -7.4013 | -86.5621 | -79.1609 | 0.1512 | 0.4623 | 0.3111 |
| | Frequency | -0.6223 | -84.2801 | -83.6578 | 0.2505 | 0.4504 | 0.1999 |
| | Pattern+ARMA | -168.4090 | -16.2990 | 152.1100 | 0.1343 | 0.6851 | 0.5507 |
| | Pattern | -143.4747 | -18.1746 | 125.3001 | 0.1536 | 0.7529 | 0.5994 |
| | ARMA | -108.4763 | -1.0564 | 107.4199 | 0.3672 | 0.6909 | 0.3237 |
| Chronos-Bolt | All | -1.1647 | 0.4096 | 1.5743 | 0.2441 | 0.8105 | 0.5664 |
| | Trend | -1.3373 | -0.2686 | 1.0686 | 0.1313 | 0.5753 | 0.4440 |
| | Frequency | -0.0674 | -0.5082 | -0.4408 | 0.1278 | 0.5974 | 0.4697 |
| | Pattern+ARMA | -1.1767 | 0.3555 | 1.5322 | 0.2370 | 0.7940 | 0.5570 |
| | Pattern | -1.1788 | 0.4547 | 1.6335 | 0.2455 | 0.8192 | 0.5737 |
| | ARMA | -0.5322 | 0.3277 | 0.8599 | 0.4677 | 0.8096 | 0.3420 |
| TimesFM | All | -10.9572 | 0.5757 | 11.5329 | 0.3429 | 0.8716 | 0.5286 |
| | Trend | -24.6562 | 0.2459 | 24.9021 | 0.1132 | 0.7616 | 0.6483 |
| | Frequency | -0.2205 | -5.3556 | -5.1351 | 0.1027 | 0.7367 | 0.6340 |
| | Pattern+ARMA | -11.0627 | 0.4657 | 11.5284 | 0.3416 | 0.8639 | 0.5223 |
| | Pattern | -10.5886 | 0.5633 | 11.1519 | 0.3410 | 0.8869 | 0.5459 |
| | ARMA | -5.3132 | 0.3740 | 5.6872 | 0.6040 | 0.8471 | 0.2431 |

Table 5: Performance comparison between full-parameter finetuning and SSIM.

| Model | Chronos | | | | | | TimesFM | | | | | |
|---|---|---|---|---|---|---|---|---|---|---|---|---|
| | Finetuning | | | SSIM | | | Finetuning | | | SSIM | | |
| Domain | $Hal \downarrow$ | $R^2 \uparrow$ | $Corr \uparrow$ | $Hal \downarrow$ | $R^2 \uparrow$ | $Corr \uparrow$ | $Hal \downarrow$ | $R^2 \uparrow$ | $Corr \uparrow$ | $Hal \downarrow$ | $R^2 \uparrow$ | $Corr \uparrow$ |
| Synthetic | 0.08 | 0.74 | 0.90 | 0.41 | 0.19 | 0.72 | 0.36 | 0.66 | 0.88 | 0.10 | 0.57 | 0.92 |
| Econ/Fin | 0.40 | -20.11 | 0.54 | 0.41 | -3.20 | 0.51 | 0.81 | -63.93 | 0.35 | 0.38 | -0.32 | 0.78 |
| Energy | 0.14 | -0.41 | 0.79 | 0.12 | 0.03 | 0.77 | 0.21 | 0.48 | 0.87 | 0.12 | 0.13 | 0.81 |
| Nature | 0.56 | -5.46 | 0.14 | 0.67 | -0.76 | 0.11 | 0.92 | -1.22 | 0.17 | 0.95 | -0.09 | 0.16 |
| Transport | 0.36 | -0.04 | 0.64 | 0.39 | -0.22 | 0.61 | 0.85 | -0.59 | 0.40 | 0.57 | 0.42 | 0.71 |
| WebOps | 0.44 | -7906.68 | 0.35 | 0.61 | -21.84 | 0.34 | 0.87 | -434686.91 | 0.17 | 0.64 | -6.75 | 0.43 |
| Aggregated | 0.33 | -1524.60 | 0.55 | 0.42 | -5.08 | 0.51 | 0.65 | -83519.74 | 0.48 | 0.43 | -1.25 | 0.64 |

# E  Additional Experimental Results

## E.1  Model Size

We examine the effectiveness of our proposed approaches on models of different sizes from the Chronos family [4]. From Table 6, SSIM effectively reduces the hallucination rate and improves the forecasting quality of different models. From Table 7, the efficacy of SSAS for hallucination detection and performance prediction is better for larger models. This is because larger models have

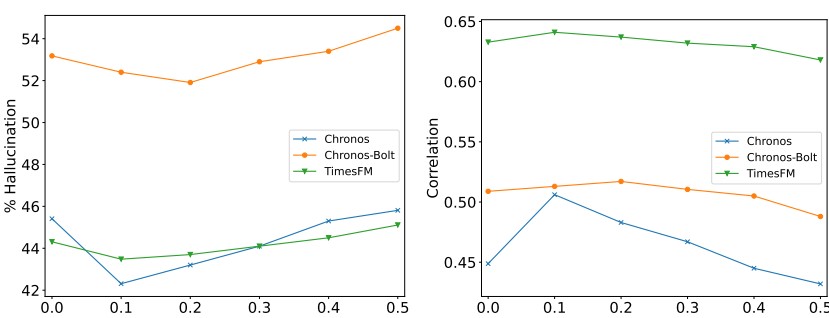

Figure 6: The aggregated mean validation performance under varying proportions of top neurons.

Table 6: Comparison of forecasting performance across domains, with the improvements boldfaced.

| Method | Domain | Chronos-Mini | | | Chronos-Small | | | Chronos-Base | | | Chronos-Large | | |
|---|---|---|---|---|---|---|---|---|---|---|---|---|---|
| | | $Hal\downarrow$ | $R^2\uparrow$ | $Corr\uparrow$ | $Hal\downarrow$ | $R^2\uparrow$ | $Corr\uparrow$ | $Hal\downarrow$ | $R^2\uparrow$ | $Corr\uparrow$ | $Hal\downarrow$ | $R^2\uparrow$ | $Corr\uparrow$ |
| Original | Synthetic | 0.6024 | -0.8781 | 0.5426 | 0.5714 | -0.7235 | 0.5623 | 0.4524 | -0.1625 | 0.6265 | 0.3048 | 0.1946 | 0.7153 |
| | Econ/Fin | 0.4307 | -2.5519 | 0.4727 | 0.4184 | -2.8915 | 0.4786 | 0.4115 | -3.3554 | 0.4751 | 0.4321 | -4.0115 | 0.4874 |
| | Energy | 0.1567 | -13.0010 | 0.6869 | 0.1309 | -9.1815 | 0.6807 | 0.1389 | -0.4839 | 0.7180 | 0.1486 | -0.2673 | 0.7055 |
| | Nature | 0.7792 | -14.3961 | 0.0613 | 0.7748 | -14.3360 | 0.0674 | 0.8035 | -10.7283 | 0.0457 | 0.7815 | -10.8791 | 0.0415 |
| | Transport | 0.4560 | -1.9068 | 0.5013 | 0.3938 | -1.5206 | 0.5415 | 0.4197 | -1.6444 | 0.5127 | 0.5130 | -1.3101 | 0.5397 |
| | WebOps | 0.6760 | -1830.7650 | 0.1665 | 0.5941 | -13.1433 | 0.2236 | 0.5801 | -414.8937 | 0.2762 | 0.5470 | -564.4352 | 0.2791 |
| | Aggregated | 0.4997 | -357.4382 | 0.4076 | 0.4665 | -7.5057 | 0.4250 | 0.4531 | -82.3762 | 0.4458 | 0.4357 | -111.1697 | 0.4604 |
| SSIM | Synthetic | **0.5095** | **-0.1507** | **0.6166** | **0.4500** | **-0.0495** | **0.6451** | **0.4145** | **0.1854** | **0.7150** | **0.2452** | **0.4594** | **0.7658** |
| | Econ/Fin | **0.4360** | **-2.0534** | **0.5522** | **0.4387** | **-2.2134** | **0.5336** | **0.4061** | **-3.2037** | **0.5146** | **0.4319** | **-1.3095** | **0.5769** |
| | Energy | **0.1373** | **-0.6871** | **0.7427** | **0.1228** | **-1.3981** | **0.7258** | **0.1191** | **0.0268** | **0.7707** | **0.1292** | **-0.5976** | **0.7543** |
| | Nature | **0.7139** | **-1.2148** | **0.0980** | **0.6894** | **-0.3257** | **0.1050** | **0.6715** | **-0.7575** | **0.1082** | **0.6960** | **-0.5360** | **0.0860** |
| | Transport | **0.4197** | **-0.1980** | **0.5880** | **0.3893** | **-0.2187** | **0.6032** | **0.3938** | **-0.2221** | **0.6081** | **0.5078** | **-0.2593** | **0.5954** |
| | WebOps | **0.6400** | **-2.2954** | **0.2413** | **0.5625** | **-1.9302** | **0.2899** | 0.6052 | **-21.8389** | **0.3369** | **0.5261** | **-1.4402** | **0.3685** |
| | Aggregated | **0.4621** | **-1.3024** | **0.4745** | **0.4320** | **-1.2709** | **0.4818** | **0.4231** | **-5.0845** | **0.5061** | **0.4051** | **-0.7534** | **0.5270** |

Table 7: The results of hallucination detection and forecasting performance prediction for TSFMs of different sizes, with the best results boldfaced. For each method, the first column shows AUROC of hallucination detection and the latter two columns show rank correlations with the performance metrics. The statistical significance of positive rank correlations is indicated with * for $p < 0.05$ and ** for $p < 0.01$.

| Model | Domain | Cosine Similarity | | | Activation Variance | | | SSAS (Ours) | | |
|---|---|---|---|---|---|---|---|---|---|---|
| | | $Hal\uparrow$ | $R^2\uparrow$ | $Corr\uparrow$ | $Hal\uparrow$ | $R^2\uparrow$ | $Corr\uparrow$ | $Hal\uparrow$ | $R^2\uparrow$ | $Corr\uparrow$ |
| Chronos-Large | Synthetic | **0.7878**$^{**}$ | 0.3553$^{**}$ | 0.2976$^{**}$ | 0.5719 | 0.1594$^{**}$ | 0.1459$^{**}$ | 0.7725 | **0.4818**$^{**}$ | **0.4905**$^{**}$ |
| | Econ/Fin | 0.8619 | 0.6584$^{**}$ | 0.6523$^{**}$ | 0.7922 | 0.6079$^{**}$ | 0.6474$^{**}$ | **0.8780** | **0.6965**$^{**}$ | **0.7055**$^{**}$ |
| | Energy | 0.6952 | **0.1309**$^{**}$ | 0.1190$^{**}$ | 0.6883 | −0.1911 | −0.0404 | **0.8190** | −0.0030 | **0.1407**$^{**}$ |
| | Nature | 0.5008 | 0.3057$^{**}$ | 0.1232$^{**}$ | 0.5020 | 0.2979$^{**}$ | 0.1239$^{**}$ | **0.5479** | **0.4150**$^{**}$ | **0.2080**$^{**}$ |
| | Transport | 0.6228 | 0.4703$^{**}$ | 0.5423$^{**}$ | 0.6994 | 0.5187$^{**}$ | 0.5728$^{**}$ | **0.7044** | **0.5523**$^{**}$ | **0.6572**$^{**}$ |
| | WebOps | **0.6121** | 0.2366$^{**}$ | **0.3800**$^{**}$ | 0.5222 | 0.2237$^{**}$ | 0.2785$^{**}$ | 0.5477 | **0.2532**$^{**}$ | 0.3525$^{**}$ |
| | Aggregated | 0.7800 | 0.5168$^{**}$ | 0.5388$^{**}$ | 0.7240 | 0.4455$^{**}$ | 0.5033$^{**}$ | **0.8035** | **0.5787**$^{**}$ | **0.6492**$^{**}$ |
| Chronos-Base | Synthetic | 0.7847 | 0.3834$^{**}$ | 0.3262$^{**}$ | 0.6786 | 0.3734$^{**}$ | 0.4052$^{**}$ | **0.8316** | **0.4299**$^{**}$ | **0.5111**$^{**}$ |
| | Econ/Fin | **0.8495** | **0.6501**$^{**}$ | **0.6208**$^{**}$ | 0.6927 | 0.5096$^{**}$ | 0.5507$^{**}$ | 0.7833 | 0.5034$^{**}$ | 0.5258$^{**}$ |
| | Energy | 0.7124 | **0.5088**$^{**}$ | **0.3528**$^{**}$ | 0.8093 | −0.1116 | 0.0373 | **0.8096** | 0.1166$^{**}$ | 0.0363 |
| | Nature | 0.4978 | 0.3382$^{**}$ | 0.1524$^{**}$ | 0.5384 | **0.3490**$^{**}$ | **0.1886**$^{**}$ | **0.5925** | 0.3430$^{**}$ | 0.1507$^{**}$ |
| | Transport | 0.6601 | 0.4706$^{**}$ | 0.5550$^{**}$ | **0.7466** | **0.5351**$^{**}$ | **0.6083**$^{**}$ | 0.6767 | 0.4158$^{**}$ | 0.5234$^{**}$ |
| | WebOps | 0.5542 | **0.2328**$^{**}$ | **0.3710**$^{**}$ | 0.5060 | 0.1363$^{**}$ | 0.2720$^{**}$ | **0.5740** | 0.1693$^{**}$ | 0.2526$^{**}$ |
| | Aggregated | 0.7903 | **0.5866**$^{**}$ | **0.5804**$^{**}$ | 0.7226 | 0.4197$^{**}$ | 0.5277$^{**}$ | **0.8086** | 0.5082$^{**}$ | 0.5758$^{**}$ |
| Chronos-Small | Synthetic | **0.7748** | 0.4697$^{**}$ | 0.3234$^{**}$ | 0.5204 | 0.4696$^{**}$ | **0.5616**$^{**}$ | 0.7367 | **0.4725**$^{**}$ | 0.4265$^{**}$ |
| | Econ/Fin | **0.8686** | **0.6334**$^{**}$ | **0.6167**$^{**}$ | 0.6337 | 0.3839$^{**}$ | 0.4081$^{**}$ | 0.6405 | 0.4002$^{**}$ | 0.4556$^{**}$ |
| | Energy | 0.6909 | **0.6957**$^{**}$ | **0.6036**$^{**}$ | 0.6000 | 0.0611 | 0.1903$^{**}$ | **0.7932** | 0.4817$^{**}$ | 0.5093$^{**}$ |
| | Nature | 0.4989 | 0.3520$^{**}$ | 0.2850$^{**}$ | 0.5236 | 0.3589$^{**}$ | **0.3672**$^{**}$ | **0.5595** | **0.3803**$^{**}$ | 0.3253$^{**}$ |
| | Transport | 0.7160 | **0.6267**$^{**}$ | 0.6033$^{**}$ | 0.7683 | 0.4515$^{**}$ | 0.6326$^{**}$ | **0.7822** | 0.5387$^{**}$ | **0.6956**$^{**}$ |
| | WebOps | **0.5695** | **0.1688**$^{**}$ | **0.2756**$^{**}$ | 0.4830 | −0.0445 | 0.1239$^{**}$ | 0.5524 | 0.0906$^{*}$ | 0.2306$^{**}$ |
| | Aggregated | **0.7733** | **0.6013**$^{**}$ | **0.5874**$^{**}$ | 0.6255 | 0.3299$^{**}$ | 0.4396$^{**}$ | 0.6876 | 0.4356$^{**}$ | 0.5408$^{**}$ |
| Chronos-Mini | Synthetic | **0.8133** | **0.4669**$^{**}$ | **0.2308**$^{**}$ | 0.3921 | −0.1445 | 0.0087 | 0.5656 | 0.1675$^{**}$ | 0.0889$^{*}$ |
| | Econ/Fin | **0.8479** | **0.6233**$^{**}$ | **0.5881**$^{**}$ | 0.2835 | −0.1790 | −0.1390 | 0.5310 | 0.2172$^{**}$ | 0.2582$^{**}$ |
| | Energy | 0.7092 | **0.6510**$^{**}$ | **0.5645**$^{**}$ | 0.6565 | −0.0651 | 0.1352$^{**}$ | **0.7568** | 0.0121 | 0.1435$^{**}$ |
| | Nature | 0.5288 | 0.3352$^{**}$ | 0.2117$^{**}$ | 0.5769 | 0.4002$^{**}$ | 0.3066$^{**}$ | **0.6528** | **0.5848**$^{**}$ | **0.4637**$^{**}$ |
| | Transport | 0.6610 | 0.4551$^{**}$ | 0.3709$^{**}$ | 0.6627 | 0.4066$^{**}$ | 0.4655$^{**}$ | **0.6817** | **0.5747**$^{**}$ | **0.6522**$^{**}$ |
| | WebOps | 0.5263 | 0.1289$^{**}$ | **0.1983**$^{**}$ | 0.5149 | **0.1870**$^{**}$ | 0.1203$^{**}$ | 0.5349 | 0.1148$^{**}$ | 0.1532$^{**}$ |
| | Aggregated | **0.7580** | **0.5664**$^{**}$ | **0.5317**$^{**}$ | 0.5855 | 0.1707$^{**}$ | 0.2853$^{**}$ | 0.6147 | 0.3091$^{**}$ | 0.3975$^{**}$ |

stronger capability of distinguishing context signals from noises through the processing of larger and more layers, leading to more distinct activation behaviors of signal neurons with respect to different types of inputs.

## E.2 No Data Filtering

We study the impact of TSFM hallucinations without filtering the time series instances using the knowledge rules in data preprocessing. We randomly sample 500 time series instances from oversized datasets to preserve data balance. From Tables 8 and 9, we note that although there is a small decline in the overall performance compared with the results with data filtering in Tables 1 and 2 of the main text, the quality of non-hallucinated forecasts is still substantially better than that of hallucinated

Table 8: Comparison of forecasting performance across domains without data filtering.

| Domain | Chronos | | | Chronos-Bolt | | | TimesFM | | |
|---|---|---|---|---|---|---|---|---|---|
| | $Hal \downarrow$ | $R^2 \uparrow$ | $Corr \uparrow$ | $Hal \downarrow$ | $R^2 \uparrow$ | $Corr \uparrow$ | $Hal \downarrow$ | $R^2 \uparrow$ | $Corr \uparrow$ |
| Synthetic | 0.4610 | -0.1701 | 0.6292 | 0.5448 | 0.0013 | 0.5544 | 0.1105 | 0.5983 | 0.9141 |
| Econ/Fin | 0.6526 | -5.5863 | 0.3556 | 0.6825 | -2.0782 | 0.4702 | 0.6937 | -1.0520 | 0.6423 |
| Energy | 0.4078 | -2.2455 | 0.5606 | 0.3643 | -0.4111 | 0.6521 | 0.3938 | -0.5814 | 0.6625 |
| Nature | 0.8736 | -2.8898 | 0.0875 | 0.9347 | -0.6486 | 0.1628 | 0.9432 | -0.2972 | 0.1918 |
| Transport | 0.7358 | -7.6617 | 0.2747 | 0.8752 | -0.8877 | 0.3657 | 0.8507 | -0.3036 | 0.4527 |
| WebOps | 0.8172 | -433.0063 | 0.1532 | 0.8616 | -1.8096 | 0.2428 | 0.8407 | -6.8100 | 0.2511 |
| Aggregated Mean | 0.6696 | -117.9305 | 0.3228 | 0.7120 | -1.2357 | 0.4055 | 0.6804 | -2.1914 | 0.4915 |

Table 9: Performance comparison of hallucinated and non-hallucinated forecasts by TSFMs without data filtering.

| Metric | Domain | Chronos | | | Chronos-Bolt | | | TimesFM | | |
|---|---|---|---|---|---|---|---|---|---|---|
| | | $Hal$ | $Non\text{-}hal$ | $Diff$ | $Hal$ | $Non\text{-}hal$ | $Diff$ | $Hal$ | $Non\text{-}hal$ | $Diff$ |
| $R^2 \uparrow$ | Synthetic | -1.1127 | 0.6360 | 1.7487 | -0.4383 | 0.5273 | 0.9656 | -1.3704 | 0.8428 | 2.2132 |
| | Econ/Fin | -7.5763 | -1.8483 | 5.7280 | -2.9005 | -0.3106 | 2.5899 | -1.5933 | 0.1737 | 1.7670 |
| | Energy | -3.5878 | -1.3212 | 2.2667 | -1.1147 | -0.0079 | 1.1069 | -1.5737 | 0.0632 | 1.6369 |
| | Nature | -2.9187 | -2.6900 | 0.2286 | -0.6573 | -0.5235 | 0.1338 | -0.3369 | 0.3629 | 0.6999 |
| | Transport | -8.4450 | -5.4807 | 2.9642 | -0.9532 | -0.4285 | 0.5247 | -0.4530 | 0.5478 | 1.0008 |
| | WebOps | -524.3276 | -24.6696 | 499.6580 | -1.3156 | -4.8857 | -3.5701 | -7.1521 | -5.0045 | 2.1476 |
| | Aggregated | -173.5672 | -5.1648 | 168.4024 | -1.4589 | -0.6837 | 0.7753 | -3.0391 | -0.3866 | 2.6524 |
| $Corr \uparrow$ | Synthetic | 0.3617 | 0.8581 | 0.4964 | 0.3568 | 0.7909 | 0.4341 | 0.7481 | 0.9347 | 0.1866 |
| | Econ/Fin | 0.1642 | 0.7151 | 0.5508 | 0.3121 | 0.8100 | 0.4980 | 0.5549 | 0.8403 | 0.2854 |
| | Energy | 0.3303 | 0.7191 | 0.3888 | 0.4583 | 0.7631 | 0.3048 | 0.4491 | 0.8012 | 0.3521 |
| | Nature | 0.0619 | 0.2643 | 0.2024 | 0.1192 | 0.7860 | 0.6668 | 0.1561 | 0.7855 | 0.6295 |
| | Transport | 0.1798 | 0.5388 | 0.3589 | 0.3090 | 0.7630 | 0.4539 | 0.3858 | 0.8336 | 0.4477 |
| | WebOps | 0.1037 | 0.3744 | 0.2706 | 0.1778 | 0.6474 | 0.4696 | 0.1750 | 0.6530 | 0.4780 |
| | Aggregated | 0.1629 | 0.6467 | 0.4838 | 0.2603 | 0.7643 | 0.5040 | 0.3375 | 0.8193 | 0.4818 |

forecasts at $p < 10^{-5}$ by unpaired $t$-tests. This demonstrates the general impact of the hallucination problem we have formulated on the forecasting performance of TSFMs.

# F    Case Study

Figures 7-9 illustrate examples of hallucinated and non-hallucinated forecasts by different TSFMs across real-world domains. The inferior performance of hallucinated forecasts is due to the TSFM failing to adequately capture the context information and generating hallucinations in the forecast that do not exist in the context, such as peaks, slopes, or patterns. The hallucinations generally deviate from the ground truths, resulting in low performance evaluation scores.

# G    Limitations

While our formulation of time series forecasting hallucinations is broadly applicable to all types of TSFMs, the proposed hallucination detection and intervention methods are applicable to white-box models only. A future direction is to include further TSFMs in our study.

# H    Impact Statement

We are the first to formulate and systematically study TSFM hallucinations to our best knowledge. We have formally defined the problem of TSFM hallucinations and outlined a set of procedures to check hallucinations in practice. We have proposed a method to identify the signal subspaces in TSFMs along with a measure to quantify the signal strength in TSFM hidden states. We have also proposed a simple and efficient intervention approach to mitigate hallucinations by magnifying the signal information in hidden states. Our work contributes to deeper understanding of TSFM trustworthiness that could foster future research in this direction.

## I  Ethics Statement

No human subjects were involved in this research, and all experiments were conducted using publicly available models and datasets, adhering to their respective licenses and use policies.

## J  Reproducibility Statement

We have provided comprehensive details of our proposed methods in §4 of the main text and Appendices A and C. The experimental setups, including model configurations, datasets, and evaluation metrics, have been thoroughly described in §5.1 of the main text and Appendix D. We utilize publicly available TSFMs and detail any modifications or specific settings used during experimentation. All datasets employed in our evaluation are standard benchmarks.

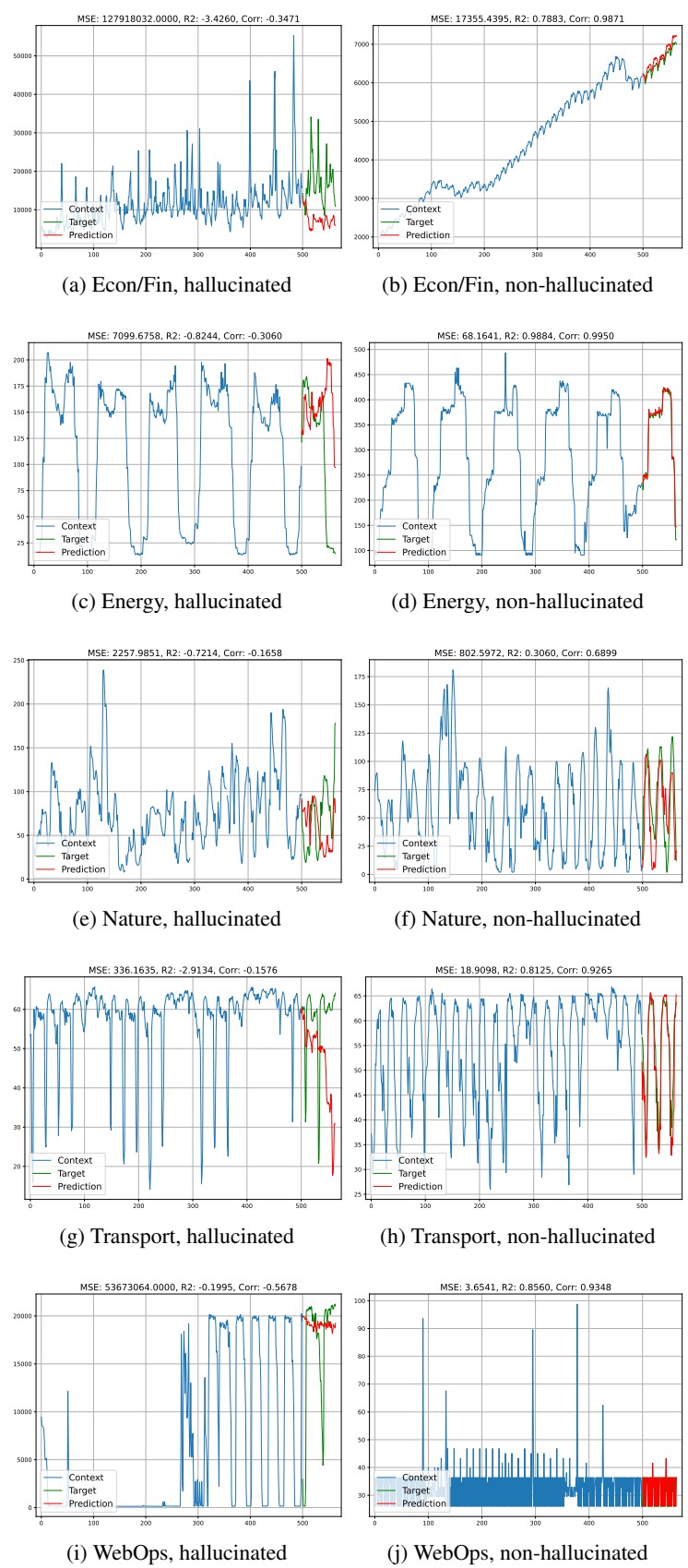

Figure 7: Examples of hallucinated and non-hallucinated forecasts by `Chronos` across domains.

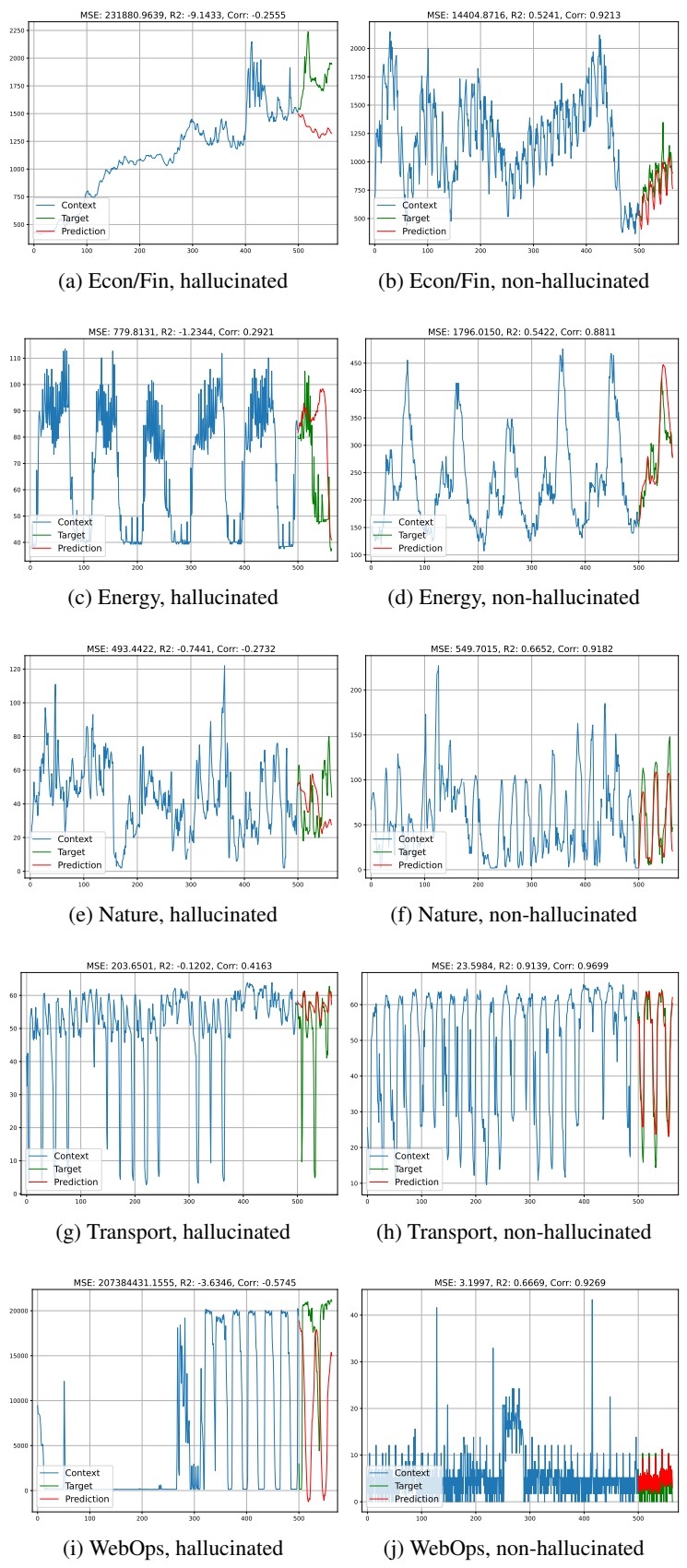

Figure 8: Examples of hallucinated and non-hallucinated forecasts by `Chronos-Bolt` across domains.

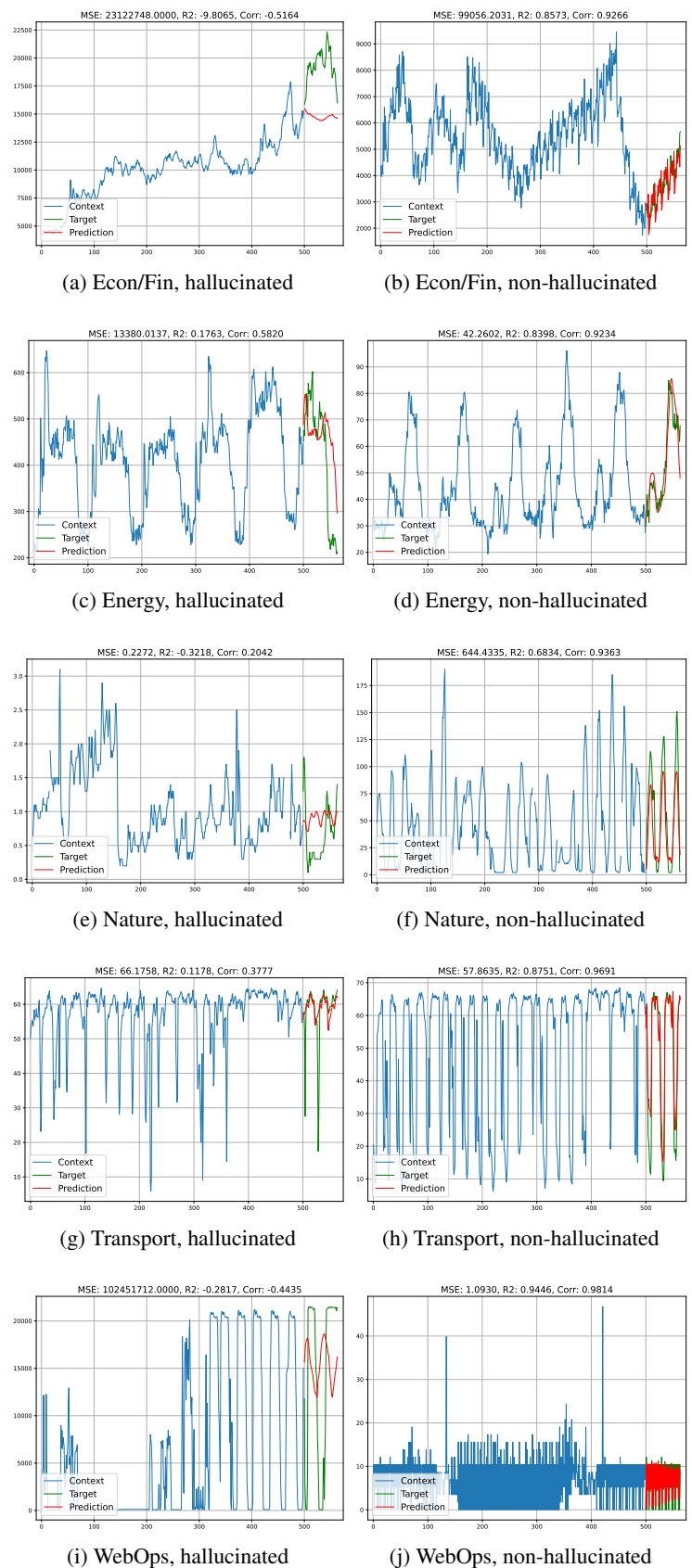

Figure 9: Examples of hallucinated and non-hallucinated forecasts by `TimesFM` across domains.

