# OpenReview forum: "Investigating Hallucinations of Time Series Foundation Models through Signal Subspace Analysis"
_NeurIPS.cc/2025/Conference — NeurIPS 2025 poster_

### Official Review · Reviewer_A1xq · 2025-06-16

**Clarity:** 3
**Significance:** 2
**Originality:** 2
**Rating:** 4
**Confidence:** 3

**Summary:**

The paper addresses hallucinations in Time Series Foundation Models (TSFMs), particularly in zero-shot forecasting settings, by defining what constitutes a hallucination based on violations of time-series dynamics inferred from the context. It proposes a novel intervention approach (SSIM) that operates on signal subspaces of hidden representations to mitigate hallucinations, and introduces a signal strength score (SSAS) to detect them. Experiments are conducted across multiple TSFMs and datasets.

**Questions:**

- How sensitive are the hallucination rules (especially ARMA/pattern-based) to scale normalization and window size?
- Have you compared SSIM against any stronger TSFM-specific debiasing or attention control methods?
- Can the proposed signal subspace activity score (SSAS) generalize to classification or anomaly detection tasks?

**Ethical Concerns:**

["NO or VERY MINOR ethics concerns only"]

**Final Justification:**

I would like to raise my score to 4 since the authors make hard works in responding most of my questions and most of their answers are satisfying. However, frankly speaking, my main concern to this paper is still not adequately addressed, which is how to correctly define "Hallucination" in the context of time series forecasting. This is because of the core difference between language and time series. Language is an abstract description to the real world, while time series are direct signals from our real world. In my understanding, the knowledge rules that the authors define are more like a set of "common errors" of "common failures". If other reviewers or the Area Chair have any concerns and would like to reject this paper, I will not oppose.

**Limitations:**

Yes

**Quality:**

2

**Strengths And Weaknesses:**

**Strengths**

- Problem Definition: The paper makes an effort to formalize the notion of hallucination in TSFMs, filling a relatively unexplored gap in the literature.

- Methodological Clarity: The intervention mechanism (CPS and SSIM) is well-described, mathematically clean, and efficient to compute at inference time.

- Empirical Evaluation: Experiments are conducted on multiple models and both synthetic and real-world datasets, showing moderate improvements over baselines in both hallucination reduction and predictive performance.

- Potential Impact: If substantiated, the idea that hidden-state subspace structure directly correlates with hallucination likelihood could provide useful insights into TSFM interpretability.

**Weakness**

- The hallucination definition relies on handcrafted rules (trend, frequency, ARMA, etc.) that are domain- and scale-sensitive, and often ill-defined or unstable in noisy, non-stationary time series.

- It is unclear whether failure on these rules always corresponds to a hallucination, or if these mismatches are natural consequences of generalization to difficult signals.

- While the CPS operation is clean, the intervention idea (projecting onto a signal subspace and magnifying) is conceptually similar to many LLM steering methods (e.g., activation steering, neuron editing).

- The main novelty seems incremental in the context of adapting these ideas to TSFMs, and prior work (e.g., [38]) has already discussed intervention in TSFM hidden spaces.

- The baseline comparisons (e.g., denoising, perturbation + averaging) are relatively weak. Stronger baselines, such as diffusion-based or fine-tuned TSFMs with dynamic attention masking, are missing.

- The correlation between SSAS and hallucination is heavily empirical; no theoretical insights are provided about why neuron variance should reliably reflect semantic grounding.

- Many insights are drawn from synthetic datasets (e.g., sinusoidal signals with additive noise), which may not reflect the complex patterns of real-world time series.

- The lack of domain-specific case studies reduces confidence in the generalizability of the findings.

---

> ### Author Rebuttal · Authors · 2025-07-31
>
> Dear Reviewer A1xq,
>
> Thank you for acknowledging the novelty of our problem definition, methodological clarity, and potential impact.
> We want to address the posed questions to improve our paper's quality and clarity. Please find the below replies to your comments.
>
> W1 The knowledge rules we propose are basic rules that capture fundamental characteristics of general time series (trend, frequency, and patterns).
> These rules are scale invariant, robust to noise, and generalizable across different domains.
> Similar TS pattern matching approaches with sliding windows have been adopted in [1].
> Experiments on real-world datasets across five different domains show that our rules effectively differentiate TSFM forecast performance (Table 2 and Appendix Table 4).
>
>
> W2 The thresholds of knowledge rules allow for a reasonable degree of generalization of context signals in TSFM forecasts.
> As mentioned in $\S5.1$, we tune the thresholds based on validation performance such that TSFM performance differentiation is maximized.
>
> W3 While our approach is intervention-based,
> the center-project-scale (CPS) operation is not used in existing works to our best knowledge.
> Our approach adaptively determines the scaling factor.
> Moreover, few existing works investigate the use of activation variance
> as the hallucination detector and forecast performance predictor.
>
> W4 As discussed in $\S2$, the work in [38] does not address specific challenges,
> such as hallucination detection and mitigation and performance prediction in our work.
>
> W5 Thanks for the suggestions. We add hallucination mitigation baselines.
> We are working on the experiments.
>
> W6 The neuron activity score (Equation 1) stems from our observations that hallucinated forecasts associate with
> low activation variance and high cosine similarity of hidden states across TS steps (Figure 1),
> while non-hallucinated forecasts with clean context signal associate with high activation variance and low cosine similarity on average (Figure 2).
> The intuition and motivation are elaborated in $\S4.1$. Essentially, activation variance reflects the sensitivity of a neuron to the context TS.
> As such, we propose to leverage the standard deviation of activations across TS steps as a measure of neuron activity.
> To distinguish neurons that are sensitive to context signal from those sensitive to context noise,
> we compute the contrastive neuron activity score (Equation 2) between data inputs without causing hallucinations and Gaussian noises.
> We select the activity score of the signal neuron with top contrastive score from the last hidden layer as the hallucination detector and forecast performance predictor, i.e., SSAS.
>
> W7 While our observations in $\S4.1$ are drawn from synthetic data,
> experiments on real-world datasets show that cosine similarity and mean activation variance alone have hallucination detection and performance prediction power to some extent (Table 3 and Appendix Table 2),
> validating the generality of observations.
> Our approach only utilizes synthetic datasets without requiring any domain-specific real-world data.
>
>
> W8 Our experiments have been conducted on real-world datasets from five different domains. We add case studies in the revision.
>
> Q1 The rules are invariant to scale as they consider the relative differences.
> Centering the context TS may have an impact on the results,
> but we do not normalize the context TS in the experiments.
> As mentioned in Appendix A, the rolling window size is always the same as the forecast horizon,
> which is necessary for computing the relative differences between the forecast and context windows.
>
> Q2 We add hallucination mitigation baselines to our evaluations.
> We are working on the experiments.
>
> Q3 We think SSAS could generalize to classification and anomaly detection tasks.
> SSAS detects hallucinations by characterizing the abnormal behaviors in hidden states
> when hallucinated forecasts are generated. Similarly, it could be useful for detecting
> whether anomalies exist in the context TS.
>
> [1] Shokoohi-Yekta, Mohammad, et al. "Discovery of meaningful rules in time series." Proceedings of the 21th ACM SIGKDD international conference on knowledge discovery and data mining. 2015.

---

> > ### Comment · Area_Chair_9B8k · 2025-08-05
> >
> > Dear reviewer,
> >
> > Thank you for your efforts so far in reviewing this paper. The authors have answered your questions.
> > Would you please check their reply and let them, the other reviewers, and myself know what you think about their response?
> >
> > Thank you,
> > Area Chair

---

> > ### Comment · Reviewer_A1xq · 2025-08-05
> >
> > I would like to raise my score to 4 since the authors make hard works in responding most of my questions and most of their answers are satisfying. However, frankly speaking, my main concern to this paper is still not adequately addressed, which is how to correctly define "Hallucination" in the context of time series forecasting. This is because of the core difference between language and time series. Language is an abstract description to the real world, while time series are direct signals from our real world. In my understanding, the knowledge rules that the authors define are more like a set of "common errors" of "common failures".

---

> ### Author Response · Authors · 2025-08-07
>
> Dear Reviewer A1xq,
>
> Thank you for raising the score. We sincerely appreciate your insightful review and feedback.
> We provide further clarification to address your concern.
> Our work focuses on the hallucinations of TSFMs in zero-shot time series forecasting.
> Analogous to LLMs processing the query and auxiliary information (such as few-shot examples) from the input context,
> TSFMs extract signal information from the context and generate extrapolations.
> A forecast is considered to be hallucinated if its dynamics drastically deviate from
> those of the context, as this suggests a failure of accurately processing the context information.
> To check if such discrepancy between the context and forecast occurs,
> we design knowledge rules to extract and compare basic dynamics of time series covering several perspectives.
> Additional ablation results on knowledge rules are provided at [Note 1](https://openreview.net/forum?id=vgfG8sEVf9&noteId=8ZwQzARiAV)
> and [Note 2](https://openreview.net/forum?id=vgfG8sEVf9&noteId=YmBPJwOyxU).
> Our work draws parallels with the works on the relationships between
> hallucinations and context information processing in the LLM domain [2, 3].
> We would be happy to answer if you have any further concern.
>
> [2] Chen, Shiqi, et al. "In-context sharpness as alerts: an inner representation perspective for hallucination mitigation." Proceedings of the 41st International Conference on Machine Learning. 2024.
>
> [3] Sun, ZhongXiang, et al. "ReDeEP: Detecting Hallucination in Retrieval-Augmented Generation via Mechanistic Interpretability." The Thirteenth International Conference on Learning Representations. 2025.

---

> > ### Comment · Reviewer_A1xq · 2025-08-08
> >
> > My understanding to hallucination is that the model outputs something that looks right but heavily violate the fact either provided by the context or in the pre-training data. Pure failure to capture the information or instruction contained in the context is closer to model’s low capacity. My concerns is that, how should we define the “looks right” for time series.

---

> ### Author Response · Authors · 2025-08-09
>
> Dear Reviewer A1xq,
>
> Thank you for your comment.
>
> The dynamics that our knowledge rules extract from the context time series can be regarded as some form of facts.
> From this perspective, our rules catch hallucinated forecasts that heavily violate the facts provided by the context.
>
> When using an LLM to solve a mathematical problem,
> people rely on external tools to verify the correctness of the LLM's output that looks right.
> The same goes for zero-shot TS forecasting.
> Since it is hard to judge whether the forecast exhibits different dynamics from
> the context with human eyes, we apply knowledge rules that involve trend, frequency, pattern, and
> ARMA analyses to check this.
> Lowering the rule tolerance thresholds $\epsilon$ would lead to a greater proportion of forecasts that look right
> being regarded as hallucinations but would sacrifice the degree of generalization in the forecasts.
>
> We believe the issue with context information processing relates to not only
> model capacity but also model pre-training,
> so it can be addressed with intervention.

---

### Official Review · Reviewer_Q2NU · 2025-07-02

**Clarity:** 3
**Significance:** 3
**Originality:** 4
**Rating:** 5
**Confidence:** 4

**Summary:**

This paper defines the hallucination of time series foundation models.  The underlying mechanisms are investigated and the signal strength of hidden states are used for checking of hallucination.  Activity score is proposed to  rank the signal neurons. Then the center-project-scale (CPS) intervention approach is proposed for mitigation of the hallucination is proposed.   It magnifies the signal information in hidden states.  Experimental results have shown the effects of the hallucination detection and mitigation.

**Questions:**

Section 4.1 describes a hallucination example and illustrates that the hidden states gives us clues about the presence of hallucinated forecasts.  It is not clear if this is just an example or if this can be generalized to most cases.   Also, are all TSFMs showing similar phenomena?  The authors should elaborate this point in order to make the proposed work convincing.

It is said that type 3 hallucinations is the most common one which include violation of the pattern and ARMA rules. It is also claimed that SSIM has the greatest impact on Type 3 hallucinations.  It is puzzling that the authors did not separate these two types of rules and analyze them individually.  As in terms of prediction, any small adjustments of ARMA coefficients may affect the prediction results.  The adjustment could be caused by the training data in the learning process.    Thus it may be fruitful to study them separately.

The legend of figure 1 may have the sub-figures (b) and (c) mixed up.

**Ethical Concerns:**

["NO or VERY MINOR ethics concerns only"]

**Limitations:**

yes

**Paper Formatting Concerns:**

NIL

**Quality:**

3

**Strengths And Weaknesses:**

The paper is clear written and well organized.  The definition of hallucination in time series based on the knowledge set is reasonable.   The paper presents the observed the patterns of the hidden states and hence form the central idea of this paper.   The only question is whether this signal and noise space of the hidden states can be generalized to all kinds of hallucination and TSFMs.  The proposed hallucination detection and hallucination mitigation methods are clearly presented.  The works are original and it may have substantial impact to future research work in this area.   The use of the knowledge set is important in this work.  Currently, trend, frequency, pattern and ARMA are considered.  It is not clear why pattern and ARMA rules are combined in this study.   Also, is it possible to include other domain knowledge in the knowledge set?

---

> ### Author Rebuttal · Authors · 2025-07-31
>
> Dear Reviewer Q2NU,
>
> Thank you for acknowledging the novelty of our work and the clarity of our paper.
> We want to address the posed questions to improve our paper's quality and clarity. Please find the below replies to your comments.
>
> W The knowledge rules we propose are basic rules that capture fundamental characteristics of general time series (trend, frequency, and patterns).
> These rules are scale invariant, robust to noise, and generalizable across different domains.
> Experiments on real-world datasets across five different domains show that our rules effectively differentiate TSFM forecast performance (Table 2 and Appendix Table 4).
> It is possible to further include domain knowledge as rules, but this may affect
> the generalizability of the knowledge set to other domains.
>
> Q1 Experiments on real-world datasets show that cosine similarity and mean activation variance alone have hallucination detection and performance prediction power to some extent (Table 3 and Appendix Table 2).
> As discussed in $\S5.2$ RQ3 and Appendix E.1, the phenomena of cosine similarity are applicable to the
> Chronos family and Chronos-Bolt. The phenomena of mean activation variance are applicable to
> all three types of models but are less consistent across domains.
>
> Q2 As explained in $\S3$, the ARMA rule is purposed to complement the pattern rule
> when the TS exhibits strong ARMA dynamics.
> Since only a small proportion of TS exhibit strong ARMA dynamics, we have not reported
> the experimental results of the standalone ARMA rule.
> We choose to use the first-order ARMA model to capture the fundamental characteristics of
> general time series and avoid overfitting.
> We are working on separate results for the pattern and ARMA rules.
>
>
> Q3 Thank you for pointing this out. We fix the typo in the revision.

---

> > ### Comment · Reviewer_Q2NU · 2025-08-07
> >
> > Thank you for your explanation.  I am looking forward to seeing the separate results.

---

> ### Author Response · Authors · 2025-08-06
> **Separate Results for Pattern and ARMA Rules**
>
> Dear Reviewer Q2NU,
>
> Thanks for your patience. Below, we report separate results for the pattern rule and ARMA rule to address **Q2** in rebuttal.
> From Table 1, both the pattern and ARMA rules contribute to the differentiation of TSFM forecast quality, with positive differentiation in all cases.
> The pattern+ARMA rule narrows down the scope of hallucinations by considering forecasts that violate both the pattern and ARMA rules as hallucinations.
> It captures a smaller set of hallucinations with poor quality and provides superior performance differentiation overall.
> Please refer to [this comment](https://openreview.net/forum?id=vgfG8sEVf9&noteId=8ZwQzARiAV) for further ablation results on knowledge rules.
> Table 2 reports the distributions of rule violations without and with SSIM intervention.
> Please note that per our definition, a forecast is also labeled as violating the ARMA rule
> when the context time series does not exhibit strong ARMA dynamics.
> SSIM primarily reduces the violations of the pattern rule.
>
>
> ---
> ### Table 1
>
> |              |      | Pattern+ARMA   Rule |          |          | Pattern Rule |          |          | ARMA Rule |         |          |
> |--------------|------|---------------------|----------|----------|--------------|----------|----------|-----------|---------|----------|
> |              |      | Hal                 | Non-hal  | **Diff**     | Hal          | Non-hal  | **Diff**     | Hal       | Non-hal | **Diff**     |
> | Chronos      | $R^2$  | -168.4090           | -16.2990 | 152.1100 | -143.4747    | -18.1746 | 125.3001 | -108.4763 | -1.0564 | 107.4199 |
> |              | Corr | 0.1343              | 0.6851   | 0.5507   | 0.1536       | 0.7529   | 0.5994   | 0.3672    | 0.6909  | 0.3237   |
> | Chronos-Bolt | $R^2$   | -1.1767             | 0.3555   | 1.5322   | -1.1788      | 0.4547   | 1.6335   | -0.5322   | 0.3277  | 0.8599   |
> |              | Corr | 0.2370              | 0.7940   | 0.5570   | 0.2455       | 0.8192   | 0.5737   | 0.4677    | 0.8096  | 0.3420   |
> | TimesFM      | $R^2$  | -11.0627            | 0.4657   | 11.5284  | -10.5886     | 0.5633   | 11.1519  | -5.3132   | 0.3740  | 5.6872   |
> |              | Corr | 0.3416              | 0.8639   | 0.5223   | 0.3410       | 0.8869   | 0.5459   | 0.6040    | 0.8471  | 0.2431   |
>
>
> ### Table 2
>
> |              | Pattern+ARMA   Rule |        | Pattern Rule |        | ARMA Rule |        |
> |--------------|---------------------|--------|--------------|--------|-----------|--------|
> |              | Original            | SSIM   | Original     | SSIM   | Original  | SSIM   |
> | Chronos      | 0.4344              | 0.413  | 0.5124       | 0.4923 | 0.7570    | 0.7567 |
> | Chronos-Bolt | 0.5100              | 0.4987 | 0.5392       | 0.5282 | 0.8765    | 0.8778 |
> | TimesFM      | 0.4347              | 0.4227 | 0.4582       | 0.4478 | 0.8651    | 0.8608 |

---

> > ### Comment · Reviewer_Q2NU · 2025-08-07
> >
> > Interesting results.  It is good to see the breakdowns as they are the most common ones.

---

### Official Review · Reviewer_43oT · 2025-07-03

**Clarity:** 3
**Significance:** 2
**Originality:** 2
**Rating:** 4
**Confidence:** 1

**Summary:**

Disclaimer: While I have experience in time series forecasting and deep learning, I do not have sufficient expertise in the specific techniques or theoretical frameworks employed in this paper (e.g., signal subspace analysis, internal representation intervention). My review may not fully reflect the technical depth of this work, and I encourage the AC to consider assigning an additional reviewer with relevant background.

Recommendation to AC: Given the specific technical depth of this work, particularly in representation manipulation and signal processing, I believe this paper would benefit from an additional review by someone with a strong background in neural representation interpretability or signal-space analysis in deep networks.

This paper investigates hallucination phenomena in Time Series Foundation Models (TSFMs). It proposes a formal definition of hallucination, introduces a method for identifying signal subspaces, and applies test-time interventions to reduce hallucination. The methods are validated on both synthetic and real-world datasets.

**Questions:**

While the motivation is clear and the empirical results appear thorough, I find it difficult to evaluate the novelty and technical correctness of the proposed intervention method and subspace decomposition approach. These appear to rely on internal mechanisms of TSFMs and assumptions that I am not qualified to fully assess.

**Ethical Concerns:**

["NO or VERY MINOR ethics concerns only"]

**Final Justification:**

N/A

**Quality:**

3

**Strengths And Weaknesses:**

While the motivation is clear and the empirical results appear thorough, I find it difficult to evaluate the novelty and technical correctness of the proposed intervention method and subspace decomposition approach. These appear to rely on internal mechanisms of TSFMs and assumptions that I am not qualified to fully assess.

---

> ### Author Rebuttal · Authors · 2025-07-31
>
> Dear Reviewer 43oT,
>
> Thank you for acknowledging the clarity of motivation and the completeness of experimental results. We appreciate your time and efforts.

---

### Official Review · Reviewer_pd2E · 2025-07-03

**Clarity:** 2
**Significance:** 2
**Originality:** 3
**Rating:** 3
**Confidence:** 5

**Summary:**

This paper explores the phenomenon of hallucinations in Time Series Foundation Models (TSFMs), introduces a formal definition in the zero-shot forecasting setting, and proposes a test-time intervention method based on identifying and amplifying signal subspaces in model hidden states.

**Questions:**

Kindly see the questions outlined in the weaknesses section.

**Ethical Concerns:**

["NO or VERY MINOR ethics concerns only"]

**Final Justification:**

The authors’ rebuttal addresses my comments only partially, and several critical concerns remain unresolved. The explanation for the HAL performance gap lacks quantitative detail on the frequency and impact of outliers; claims of statistical significance are not supported by p-values, effect sizes, or per-domain analysis; the “pattern frequency” concept remains undefined in operational terms and is unaccompanied by quantitative evidence; and the metrics section, while clearer, still omits important specifics and methodological justifications. These shortcomings reduce confidence in the validity and robustness of the reported results. Given the absence of sufficient new evidence to address these core methodological issues, **I maintain my score at a borderline reject.**

**Limitations:**

This work has some limitations. It focuses on zero-shot forecasting, leaving other probing and finetuning unexplored. Identifying signal subspaces may not generalize to all TSFMs, and the signal strength measure needs further interpretation. The intervention adds test-time overhead, and the evaluation criteria might not cover all types of forecast errors in practice.

**Paper Formatting Concerns:**

No major formatting issues observed. The paper appears to follow the NeurIPS 2025 formatting guidelines appropriately, including structure, references, figures, and overall layout.

**Quality:**

2

**Strengths And Weaknesses:**

**Strengths**

- Timely Topic: Hallucination analysis in TSFMs is underexplored. Drawing parallels with hallucinations in NLP and vision models is meaningful and relevant.
- Insightful Hypothesis: The idea that hallucinations stem from diminished signal representation in hidden states is plausible and interesting.
- Signal Subspace Analysis and Intervention Strategy:  Proposing a Signal Subspace Activity Score (SSAS) and identifying "signal neurons" using contrastive analysis adds a fresh angle to understanding hidden states in TSFMs. The proposed test-time intervention (via signal subspace magnification) is simple and avoids retraining, which is attractive for practical deployments.

**Weaknesses**

1. Literature Positioning & Missing Related Work
 - There's insufficient discussion on similar latent space analyses in time series models or comparable intervention techniques in other domains like disentanglement for time series

2. Framing and Definitions
 - The framing of hallucination is imprecise and overly broad in places. For example, defining hallucinations strictly based on deviations from handcrafted rules (trend, frequency, ARMA) is fragile and may not generalize across domains.
- Definitions (esp. Definition 2 and 3) are complicated and sometimes tautological,  hallucinations are defined via violations of rules, but those rules are subjective and can depend on thresholds chosen arbitrarily.

3. Unclear Methodology
 - The contrastive neuron activity score is underdeveloped theoretically. Why this specific measure? Why should variance across TS steps signify "signal" rather than something else (e.g., noise sensitivity or instability)?
 - The intervention mechanism remains vague: What exact operation is applied to hidden states during test-time intervention?

4. Experimental Validation
 - a) The experiments lack rigorous baselines. There's no comparison to other hallucination mitigation strategies or any ablation on how each rule contributes to hallucination detection.
 - b) The evaluation of the proposed SSAS metric as a predictor for hallucinations and forecast quality is unconvincing. Could the authors clarify under which conditions SSIM might fail or be less effective, especially given the varying results of baseline methods? Additionally, how does SSIM’s performance scale with model complexity and computational overhead compared to perturbation averaging?
 - c) There is a lack of variation analysis across different seeds for each method, such as Denoising and Perturbation+Averaging. These methods may be sensitive to hyperparameter settings.

5. Clarity & Presentation
 - a) Clarity and Organization: The paper is cluttered and difficult to follow in several sections. The overall presentation would benefit from clearer structure and more focused explanations.

 - b) Figures Not Well Integrated: Many figures, such as Figure 1, are not clearly explained or referenced in the main text. For example, it’s unclear what exactly is being visualized on the right-hand side of Figure 1, and how it supports the authors' claims.

 - c) Cosine Similarity Ambiguity: It is not clear what the cosine similarity measures in this context. Is it computed between embeddings, hidden states, or the projection layer weights? This distinction is important for interpreting the results.

 - d) Gap in Activation Statistics: The difference in standard deviation between hallucinated and non-hallucinated forecasts appears small. Why is such a narrow statistical gap sufficient to explain the shift from hallucination to accurate forecasting?

 - e) Interpretation of Representation Alignment: The authors seem to suggest that when representations are aligned in a certain direction, hallucinations are less likely. If this is the case, it could be an interesting and valuable insight for the community. However, this idea is currently underexplained and would benefit from a more rigorous and intuitive justification.

 - f) UMAP Projection: The direction in the UMAP projection (Figure 1, right) corresponding to models that do not hallucinate is not clearly defined. Without a clear definition, it is difficult to assess the significance of the observed alignment and determine whether it meaningfully correlates with hallucination behavior.

---

> ### Author Rebuttal · Authors · 2025-07-31
>
> Dear Reviewer pd2E,
>
> Thank you for acknowledging the importance of our research topic, novelty of our methodology, and potential impact.
> We want to address the posed questions to improve our paper's quality and clarity. Please find the below replies to your comments.
>
> W1 We have discussed on [38] in the paper.
> We add discussions on [1,2,3]. If there are additional works, we would appreciate pointing them out.
>
> W2 The knowledge rules we propose are basic rules that capture fundamental characteristics of general time series (trend, frequency, and patterns).
> These rules are scale invariant, robust to noise, and generalizable across different domains.
> Similar TS pattern matching approaches with sliding windows have been adopted in [4].
> Experiments on real-world datasets across five different domains show that our rules effectively differentiate TSFM forecast performance (Table 2 and Appendix Table 4).
> The thresholds of rules allow for a reasonable degree of generalization of context signals in TSFM forecasts.
> As mentioned in $\S5.1$, the thresholds are tuned based on validation performance such that TSFM performance differentiation is maximized.
>
> W3 The neuron activity score (Equation 1) stems from our observations that hallucinated forecasts associate with
> low activation variance and high cosine similarity of hidden states across TS steps (Figure 1),
> while non-hallucinated forecasts with clean context signal associate with high activation variance and low cosine similarity on average (Figure 2).
> The intuition and motivation are elaborated in $\S4.1$. Essentially, activation variance reflects the sensitivity of a neuron to the context TS.
> As such, we propose to leverage the standard deviation of activations across TS steps as a measure of neuron activity.
> To distinguish neurons that are sensitive to context signal from those sensitive to context noise,
> we compute the contrastive neuron activity score (Equation 2) between data inputs without causing hallucinations and Gaussian noises.
> We select the activity score of the signal neuron with top contrastive score from the last hidden layer as the hallucination detector and forecast performance predictor, i.e., SSAS.
> Experiments on real-world datasets show that while cosine similarity and mean activation variance alone have hallucination detection and performance prediction power to some extent,
> SSAS yields superior performance by considering the activity of identified signal neurons (Table 3 and Appendix Table 2).
>
> In SSIM, we scale the activations of candidate signal neurons via our proposed Center-Project-Scale (CPS) operation described in $\S4.3$.
> The properties of CPS operation are mathematically proved in Appendix B. The algorithm of SSIM is outlined in Appendix C.
>
> W4 a) We add hallucination mitigation baselines.
> We add an ablation study on how each rule contributes to hallucination detection.
> We are working on these experiments.
>
> W4 b) SSIM could fail when the number of pattern occurrences in the context is low.
> It is easier for TSFMs to capture frequent patterns than rare patterns from the context TS.
>
> The comparison of SSIM performance for models of different complexity is presented in Appendix E.1.
> The computational overhead of SSIM at each layer is $O(nk)$, where $n$ is the number of TS steps and $k$ is the number of signal neurons.
> The computational cost of perturbation averaging is the number of perturbations times the original computational cost of TSFM inference.
>
> W4 c) Denoising with sliding windows does not involve randomization. We set the window size to 5 as it works the best overall.
> In Perturbation+Averaging, we perturb the input TS with Gaussian noise for 10 runs with different random seeds and average the outputs.
>
> W5 b) Figure 1 (c) and (d) compare the distributions of last layer hidden states
> corresponding to the hallucinated forecast in Figure 1 (a) and non-hallucinated forecast in Figure 1 (b).
> We observe the hidden states in (d) are more evenly distributed within each cluster than those in (c).
> Figure 1 has been referenced in $\S1$ and $\S4.1$ main text.
> We note that the subfigures should instead be listed as (a) (b) ... (c) (d) ... in Figure 1 caption.
>
> W5 c) We compute the mean pairwise cosine similarity of hidden states across TS steps,
> which has also been used as a measure in [5].
>
> W5 d) We assume the activation statistics in Figure 1 (c) (d) are being referred to in the comment.
> These show the difference of the mean activation standard deviation of all neurons.
> By identifying the signal neurons, SSAS yields greater gaps
> between hallucinated and non-hallucinated forecasts with substantially higher hallucination detection and
> performance prediction accuracies than mean activation standard deviation (Table 3 and Appendix Table 2)
>
>
> W5 e) Our observations suggest that hallucinated forecasts associate with
> low activation variance and high cosine similarity of hidden states across TS steps (Figure 1),
> while non-hallucinated forecasts with clean context signal associate with high activation variance and low cosine similarity on average (Figure 2).
> TSFM hallucinations are partly caused by the inactivity of signal neurons, leading to more similar hidden states.
> Experiments on real-world datasets show that cosine similarity and mean activation variance of hidden states have hallucination detection and performance prediction power to some extent (Table 3 and Appendix Table 2).
> We mathematically show in Appendix Theorem 3 that our CPS intervention operation
> effectively reduces the cosine similarity of hidden states that have different projection directions in the signal subspace.
>
> W5 f) Through UMAP projection, we aim to show in Figure 1 (c) (d) the difference in the distributional characteristics of last layer hidden states corresponding to the hallucinated forecast in Figure 1 (a) and non-hallucinated forecast in Figure 1 (b)
> rather than alignment to certain directions.
> We observe the hidden states in (d) are more evenly distributed within each cluster than those in (c).
>
>
> [1] Santamaria-Valenzuela, Inmaculada, et al. "Decoding Latent Spaces: Assessing the Interpretability of Time Series Foundation Models for Visual Analytics." arXiv preprint arXiv:2504.20099 (2025).
>
> [2] Li, Yuening, et al. "Towards learning disentangled representations for time series." Proceedings of the 28th ACM SIGKDD Conference on Knowledge Discovery and Data Mining. 2022.
>
> [3] Oublal, Khalid, et al. "Disentangling time series representations via contrastive independence-of-support on l-variational inference." The Twelfth International Conference on Learning Representations. 2024.
>
> [4] Shokoohi-Yekta, Mohammad, et al. "Discovery of meaningful rules in time series." Proceedings of the 21th ACM SIGKDD international conference on knowledge discovery and data mining. 2015.
>
> [5] Nguyen, Tam, Tan Nguyen, and Richard Baraniuk. "Mitigating over-smoothing in transformers via regularized nonlocal functionals." Advances in Neural Information Processing Systems 36 (2023): 80233-80256.

---

> > ### Author Response · Authors · 2025-08-08
> > **Gap in Activation Statistics**
> >
> > Dear Reviewer pd2E,
> >
> > We report activation statistics to further address **W5 d)** in rebuttal.
> > The table below compares the mean SSAS versus mean activation standard deviation from our experiments.
> > We observe that mean activation std has hallucination detection power to some extent.
> > By identifying the signal subspace as detailed in $\S4.2$,
> > SSAS yields substantially greater gaps between hallucinations and non-hallucinations,
> > explaining its superior performance in Table 3 over baseline statistics.
> > The activation statistics (mean activation std) shown in Figure 1 (c) (d) are mainly purposed for building the intuition.
> >
> >
> >
> > |              | SSAS          |                    |            | Mean   Activation Std |                    |            |
> > |--------------|---------------|--------------------|------------|-----------------------|--------------------|------------|
> > |              | Hallucination |  Non-hallucination | Difference | Hallucination         |  Non-hallucination | Difference |
> > | Chronos      | 38.7247       | 54.6944            | 15.9697    | 13.1977               | 14.8077            | 1.6100     |
> > | Chronos-Bolt | 1.3049        | 1.6369             | 0.3320     | 0.5001                | 0.5205             | 0.0204     |
> > | TimesFM      | 24.8238       | 29.7494            | 4.9256     | 4.3334                | 1.4110             | -2.9224    |

---

> ### Author Response · Authors · 2025-08-05
> **Rule Ablations**
>
> Dear Reviewer pd2E,
>
> Thanks for your patience. Below, we report the ablation results on the effect of each knowledge rule that constitutes our definition of TS forecast hallucinations to address **W4 a)** in rebuttal.
> The distributions of rule violations have been provided in Figure 4.
> From the table below, we note that all rules contribute to the differentiation of forecast quality, with the aggregated Pearson correlation of non-hallucinations substantially higher than that of hallucinations in all cases.
> The pattern+ARMA rule effectively differentiates both $R^2$ and correlations in all cases.
> The frequency rule differentiates correlations but not $R^2$, due to the misalignment between forecasts and ground truths on some outlier test instances.
> The synergy of the three rules gives the best performance differentiation overall.
>
>
>
> |              |      | All Rules |          |          | Trend Rule |          |          | Frequency Rule |          |          | Pattern+ARMA   Rule |          |          |
> |--------------|------|-----------|----------|----------|------------|----------|----------|----------------|----------|----------|---------------------|----------|----------|
> |              |      | Hal       | Non-hal  | **Diff** | Hal        | Non-hal  | **Diff** | Hal            | Non-hal  | **Diff** | Hal                 | Non-hal  | **Diff** |
> | Chronos      | $R^2$   | -161.6823 | -16.6599 | 145.0224 | -7.4013    | -86.5621 | -79.1609 | -0.6223        | -84.2801 | -83.6578 | -168.4090           | -16.2990 | 152.1100 |
> |              | Corr | 0.1459    | 0.6943   | 0.5484   | 0.1512     | 0.4623   | 0.3111   | 0.2505         | 0.4504   | 0.1999   | 0.1343              | 0.6851   | 0.5507   |
> | Chronos-Bolt | $R^2$   | -1.1647   | 0.4096   | 1.5743   | -1.3373    | -0.2686  | 1.0686   | -0.0674        | -0.5082  | -0.4408  | -1.1767             | 0.3555   | 1.5322   |
> |              | Corr | 0.2441    | 0.8105   | 0.5664   | 0.1313     | 0.5753   | 0.4440   | 0.1278         | 0.5974   | 0.4697   | 0.2370              | 0.7940   | 0.5570   |
> | TimesFM      | $R^2$   | -10.9572  | 0.5757   | 11.5329  | -24.6562   | 0.2459   | 24.9021  | -0.2205        | -5.3556  | -5.1351  | -11.0627            | 0.4657   | 11.5284  |
> |              | Corr | 0.3429    | 0.8716   | 0.5286   | 0.1132     | 0.7616   | 0.6483   | 0.1027         | 0.7367   | 0.6340   | 0.3416              | 0.8639   | 0.5223   |

---

> > ### Comment · Reviewer_pd2E · 2025-08-08
> >
> > Thank you to the authors for their responses. However, the new experiment related to my comment **W.4(a)** raises further confusion. It is unclear where such a large gap in **HAL** performance comes from, for example, Chronos shows an $R^2$ of -161.68, while Chronos+BoLST yields -1.16. These results seem highly inconsistent and unexpected.
> >
> > Moreover, as noted in my initial review, the performance difference between HAL and non-HAL variants remains marginal and not clearly justified. Regarding point **W.4(b) and W.5**, the authors’ claim that “it is easier for TSFMs to capture frequent patterns than rare patterns” remains unconvincing. Most TSFM-based models underperform on various benchmarks when sequence is large, and making such a strong claim without solid evidence is problematic. As for **W.4(c)**, the authors chose a window size of 5, which seems too small to support general conclusions.
> >
> > Overall, the evaluation still lacks clarity, the metrics section needs significant restructuring, and the experimental support for the authors' claims is still insufficient. The paper remains in need of serious restructuring, and thus I find no justification for adjusting my score.

---

> ### Author Response · Authors · 2025-08-09
>
> Dear Reviewer pd2E,
>
> Thank you for your comment. We provide further clarification to address your concern.
>
> > It is unclear where such a large gap in HAL performance comes from.
>
> The additional rule ablation results are consistent with Table 2 in the paper.
> We find that the huge negative $R^2$ scores come from a few outlier test instances.
> On domains with large peaks, such as WebOps,
> Chronos sometimes generates a forecast with the peak misaligned with the ground truth,
> resulting in a huge negative $R^2$ score.
> Since Chronos-Bolt and TimesFM adopt patching on the input, they
> generate smoother forecasts than Chronos,
> so their results are less affected by extreme $R^2$ scores.
> You may also refer to the correlation results for clearer performance comparison
> as Pearson correlation coefficients range within $[-1,1]$ without yielding extreme scores.
>
> > The performance difference between HAL and non-HAL variants remains marginal and not clearly justified.
>
> We assume the performance differences in Table 2 in the paper and the Rule Ablation table are being referred to in the comment.
> In these tables, $R^2$ ranges within $(-\infty,1]$, and Pearson correlation coefficient ranges within $[-1,1]$.
> The Diff column shows that there is a significant gap of $R^2$ and Pearson correlation between HAL and non-HAL forecasts,
> as validated by the $\chi^2$ test.
> The inferior performance of hallucinated forecasts is because
> the TSFM fails to accurately process the context information and
> generates hallucinations in the forecast that do not ever exist in the context,
> such as a peak, slope, or pattern.
> These hallucinations generally deviate from the ground truths, resulting in low evaluation scores.
>
> > The authors’ claim that “it is easier for TSFMs to capture frequent patterns than rare patterns” remains unconvincing.
>
> In this claim, we refer to the number of pattern occurrences within a fixed length of input context.
> For instance, $sin(2x)$ has a higher frequency than $sin(x)$,
> and it is easier for TSFMs to capture the patterns of $sin(2x)$.
> On real-world datasets, we observe that TSFMs generally have better performance when the patterns are frequent and regular,
> such as the ones from the Energy domain in our experiments.
>
> > The metrics section needs significant restructuring.
>
> We apologize for the confusion. We revise the metrics section as follows:
>
> We evaluate forecast quality with $R^2$ and Pearson correlation coefficient, which are scale invariant. $R^2$ measures the goodness of fit to the ground truth;
> Pearson correlation coefficient measures the strength and direction of the linear relationship with the ground truth (invalid values are imputed with 0). Whether a forecast is hallucinated is determined according to the knowledge rules defined in $\S3$. We evaluate the effect of hallucination
> mitigation with hallucination rate reduction and forecast quality improvement. We evaluate the accuracy of hallucination detection with AUROC and performance prediction with Spearman rank correlation coefficient.
>
> Would this help?

---

### Comment · Reviewer_pd2E · 2025-10-30
**Promised Revisions Not Addressed at All**

It is disappointing to see that the camera-ready version of the paper does not address the issues the authors explicitly promised to resolve during the review process. This lack of follow-through is concerning, as these points were raised by reviewers and highlighted by the meta-reviewer.

While the paper has been accepted, it is essential that the authors take these comments seriously and ensure they are properly addressed in the final version. Ignoring reviewer concerns undermines the integrity of the review process and the quality of the publication.

I strongly urge the authors to revisit the reviewers’ and meta-reviewer’s comments before the final submission.

---

### Comment · Reviewer_A1xq · 2025-10-30
**Concern for the camera ready version**

As pointed out by reviewer pd2E, the concerns are not fully addressed by the authors in the camera-ready version as promised. I am also deeply concerned for this.

---

> ### Comment · Reviewer_A1xq · 2025-10-31
> **Missing Revisions**
>
> I am not enumerating all missing revisions because I have not gotten a chance to do it until today (Reviewer pd2E's comment reminded me to do so). Here are some missing revisions involved in the rebuttal to my review:
> 1. In the response to W8. The authors reply that "We add case studies in the revision". However, I did not find any new case study in the camera-ready version. The only case study that I saw is the Figure 1, which seems not to be a real-world data case and already included in the original version
> 2. In the response to my W5: "Thanks for the suggestions. We add hallucination mitigation baselines. We are working on the experiments." I did not see the new hallucination mitigation baselines in the camera-ready revision, neither following my suggestion to use some diffusion based model to post-process the output or following some recent hallucination-related works in NLP area. The "full-parameter finetune (FT) TSFMs" in the rebuttal is not included in the paper, neither.
> 3. I did not see the Note 1 and 2 ablation study on different rule combinations (they put the hyper links to the rebuttal to reviewer pd2E) in the camera-ready version.

---

> > ### Public Comment · ~Yufeng_Zou1 · 2025-10-31
> >
> > Dear Reviewer,
> >
> > Thank you for your message and for flagging the critical discrepancy in our camera-ready submission.
> > We were disturbed to see your comment and immediately investigated the issue.
> > We have identified a critical mistake in our submission process.
> > We must clarify that all the promised revisions, including the "full-parameter finetune (FT) TSFMs" and case study results, were indeed completed before the camera-ready deadline.
> > However, in the final chaotic rush of submission, we made a version-control error and incorrectly uploaded an incomplete/older version of our manuscript and supplementary materials.
> > This was a terrible procedural error on our part. The correct, complete version of the paper and supplementary (which we intended to submit) is [presented here](https://drive.google.com/file/d/1pnv3vUpsRg8aTVeymy6MhII_4QLu0wWm/view?usp=sharing) for your immediate inspection.
> > We will contact the relevant parties if it is possible to submit the correct version despite being past the due date.
> > We sincerely apologize for this mistake. There was not the slightest intent to not perform all of the tasks promised. We appreciate your understanding.

---

### Note · Authors · 2025-08-12

Dear Reviewers,

Thank you again for your insightful review and feedback. We sincerely appreciate your time and efforts. Below, we provide additional responses to your comments.

**Reviewer pd2E**
> As for W.4(c), the authors chose a window size of 5, which seems too small to support general conclusions.

As mentioned in the rebuttal, we set the window size of the denoising baseline to 5 based on validation performance.
A smaller window size would reduce the denoising effect, while a larger window size would over smooth the context input and cause signal loss, both deteriorating the performance.

**Reviewers pd2E & A1xq**
> Stronger baselines.

As a stronger hallucination mitigation baseline, we full-parameter finetune (FT) TSFMs using the validation data of each domain with 1,000 training steps.
The table below compares the forecasting performance across domains.
We observe that the performance of SSIM for Chronos is on par with full-parameter finetuning on most domains and
better on some domains such as Energy, while finetuning suffers poor performance on some test instances due to overfitting.
For TimesFM, finetuning suffers from even more severe overfitting, leading to inferior performance on all domains except Energy.
Energy is a domain with frequent and regular patterns. SSIM saves the overheads of finetuning and better preserves the generalization performance of TSFMs.

| | Chronos   | | | | | | TimesFM | | | | | |
|-|-|-|-|-|-|-|-|-|-|-|-|-|
| | FT| | | SSIM | | | FT| | | SSIM | | |
| Domain | Hal | $R^2$ | Corr | Hal  | $R^2$ | Corr | Hal | $R^2$ | Corr | Hal  | $R^2$ | Corr |
| Synthetic | 0.08 | 0.74 | 0.90 | 0.41 | 0.19 | 0.72 | 0.36 | 0.66 | 0.88 | 0.10 | 0.57 | 0.92 |
| Econ/Fin | 0.40 | -20.11 | 0.54 | 0.41 | -3.20  | 0.51 | 0.81 | -63.93 | 0.35 | 0.38 | -0.32 | 0.78 |
| Energy | 0.14 | -0.41 | 0.79 | 0.12 | 0.03 | 0.77 | 0.21 | 0.48 | 0.87 | 0.12 | 0.13 | 0.81 |
| Nature | 0.56 | -5.46 | 0.14 | 0.67 | -0.76 | 0.11 | 0.92 | -1.22 | 0.17 | 0.95 | -0.09 | 0.16 |
| Transport | 0.36 | -0.04 | 0.64 | 0.39 | -0.22 | 0.61 | 0.85 | -0.59 | 0.40 | 0.57 | 0.42 | 0.71 |
| WebOps | 0.44 | -7906.68 | 0.35 | 0.61 | -21.84 | 0.34 | 0.87 | -434686.91 | 0.17 | 0.64 | -6.75 | 0.43 |
| Aggregated Mean | 0.33 | -1524.60 | 0.55 | 0.42 | -5.08 | 0.51 | 0.65 | -83519.74 | 0.48 | 0.43 | -1.25 | 0.64 |

---

### Decision · Program_Chairs · 2025-09-17

**Decision:**

Accept (poster)

**Comment:**

The paper studies hallucinations in time series foundation models, especially in the zero-shot setting. It suggests a method to detect hallucination (SSAS) and another method to mitigate hallucination (SSIM).

The hallucination is defined as a change in certain statistics of the data between the context given to the foundation model and the predicted output. These statistics, or knowledge set as the paper calls them, are based on the Trends, Frequency, a certain definition of pattern, and the fit of an ARMA model.


We have a wide range of opinions about this paper. There is one accept (5), two borderline accepts (4), and one borderline reject (3). One of the borderline accepts is of low confidence, so we effectively have scores of 3, 4, 5 with high confidence.


The positive aspects of the paper is the novelty and originality of the method, which may in the future lead to better and more effective methods.

There are some negative aspects too:

The concern of Reviewer A1xq (Score: 4) is how to correctly define Hallucination in the context of time series, and whether the proposed criteria is the correct on. On the other hand, Reviewer Q2NU (score: 5) believes that the definition is reasonable.

In my own reading of the paper, I tend to agree that the definition is a bit arbitrary, as it depends on how the "knowledge set" is defined. Someone else could choose a different knowledge sets with different predictions potentially being detected as hallucination. This does not, however, invalidate the whole framework, as the proposed methods would still be applicable, though with different results.

There was a concern (Reviewers pd2E and A1xq) about missing baselines. The authors have added new results, based on model fine-tuning, during the rebuttal period, and at least partially addressed it.

Another concern is related to the Statistical Significance of the results, which is raised by Reviewer pd2E. While the authors perform significance tests for some of the results, some others lack any statistical significance. For example, in Figure 4, the proportion of hallucinations - type 3 for Chronos-Bolt is reduced from 51.00 to 49.87, and it is reduced from 43.44 to 41.30 in Chronos. These numbers are very close to each other and may not even be statistically different.

Another issue is the low R2 score for almost all methods, baselines and the introduced SSIM, as they often range in the negative values – this is worse than a very simple estimator that predicts the average value. Yet, the proposed method is better than the alternative baselines. This is perhaps more an issue with the foundation models for time-series prediction than the proposed method. During the rebuttal, the authors attributes very negative R2 scores to a few outlier test instances. Reviewer pd2E is not fully satisfied.

Overall, I believe the novelty of the paper and the potential impact of the framework might overcome its weaknesses.